# Iteratively Learning Novel Strategies with Diversity Measured in State Distances

## Abstract

In complex reinforcement learning (RL) problems, policies with similar rewards may have substantially different behaviors. Yet, to not only optimize rewards but also discover as many diverse strategies as possible remains a challenging problem. A natural approach to this task is constrained population-based training (PBT), which simultaneously learns a collection of policies subject to diversity constraints. However, due to the unaffordable computation cost of PBT, we adopt an alternative approach, iterative learning (IL), which repeatedly learns a single novel policy that is sufficiently different from previous ones. We first analyze these two frameworks and prove that, for any policy pool derived by PBT, we can always use IL to obtain another policy pool of the same rewards and competitive diversity scores. In addition, we also present a novel state-based diversity measure with two tractable realizations. Such a metric can impose a stronger and much smoother diversity constraint than existing action-based metrics. Combining IL and the state-based diversity measure, we develop a powerful diversity-driven RL algorithm, *State-based Intrinsic-reward Policy Optimization* (SIPO), with provable convergence properties. We empirically examine our algorithm in complex multi-agent environments including StarCraft Multi-Agent Challenge and Google Research Football. In these environments, SIPO is able to consistently derive strategically diverse and human-interpretable policies that cannot be discovered by existing baselines.

## 1 Introduction

A consensus in deep learning (DL) is that most local optima have similar losses to the global optimum (Venturi et al., 2018; Roughgarden, 2020; Ma, 2021). Hence, via stochastic gradient descent (SGD), most DL works only focus on the final performance of the learned model without considering *which* local optimum SGD discovers. However, such a performance-oriented paradigm can be problematic for reinforcement learning (RL) because it is typical in complex RL problems that policies with the same reward may have substantially different behaviors. For example, a high-reward agent in a boat-driving game can either carefully drive the boat or keep turning around to exploit an environment bug (Clark & Amodei, 2016); a humanoid football AI can adopt any dribbling or shooting behaviors to score a goal (Liu et al., 2022); a strong StarCraft AI can take very distinct construction and attacking strategies (Vinyals et al., 2019). Thus, it is a fundamental problem for an RL algorithm to not only optimize rewards but also discover as many diverse strategies as possible.

In order to obtain diverse RL strategies, we can naturally extend single-policy learning to population-based training (PBT). The problem can be formulated as a constrained optimization problem by simultaneously learning a collection of policies subject to policy diversity constraints (Parker-Holder et al., 2020b; Lupu et al., 2021). However, since multiple policies are jointly optimized, PBT can be computationally challenging (Omidshafiei et al., 2020).Therefore, a greedy alternative is *iterative learning*, which iteratively learns a single novel policy that is sufficiently different from previous ones (Masood & Doshi-Velez, 2019; Zhou et al., 2022). Since only one policy is learned per iteration, IL can largely simplify optimization. However, there have not been any theoretical guarantees on the performance or the convergence properties of IL methods.

In addition to the computation frameworks, how to quantitatively measure the difference (i.e., *diversity*) between two policies remains an open question as well. Mutual information (MI) is perhaps

the most popular diversity measure (Eysenbach et al., 2019). Although MI reveals great potential to discover diverse locomotion skills, it is proved in Eysenbach et al. (2022) that maximizing MI will not recover the set of optimal policies w.r.t. the environment reward. Therefore, MI-based methods often serve as a pre-training phase for downstream tasks (Sharma et al., 2020; Campos et al., 2020). Another category of diversity measure is based on the action distributions, such as Wasserstein distance (Sun et al., 2020), cross-entropy (Zhou et al., 2022), and Jensen-Shannon divergence (Lupu et al., 2021). Action-based measures are straightforward to evaluate and optimize. However, we will show in Sec. 4.2 that such a metric can completely fail in simple scenarios.

In this paper, we present comprehensive studies to address the two issues above. First, we provide an in-depth analysis of the two computation frameworks, namely PBT and IL, for learning diverse strategies. We theoretically prove that, in addition to simplified optimization thanks to fewer constraints, IL can discover solutions with the same reward as PBT with *at least* half of the diversity score. Regarding the diversity measure, we consider two concrete scenarios, i.e., grid-world navigation and Google Research Football (GRF). In the grid-world example, we construct visually different strategies that cannot be distinguished by popular action-based diversity measures. In the GRF example, we show that duplicated actions taken by an idle player can drastically influence the action-based diversity score. Consequently, we argue that an effective diversity measure should focus on state distances instead of action distributions.

Combining IL and a state-based diversity measure, we design a generic and effective algorithm, *State-based Intrinsic-reward Policy Optimization (SIPO)*, for discovering diverse RL strategies in an iterative fashion. In each iteration, SIPO learns a single novel policy with state-based diversity constraints w.r.t. policies learned in previous iterations. We further solve this constrained optimization problem via Lagrangian method and two-timescale gradient descent ascent (GDA) (Lin et al., 2020). Theoretical results show that our algorithm is guaranteed to converge to a neighbour of $\epsilon$-approximate KKT point (Dutta et al., 2013). Regarding the state-based measure, we provide two practical realizations, including a straightforward version based on the RBF kernel and a more general learning-based variant using Wasserstein distance.

We validate the effectiveness of our algorithm in two challenging multi-agent environments: Star-Craft Multi-Agent Challenge (Samvelyan et al., 2019) and Google Research Football (Kurach et al., 2020). Specifically, our algorithm can successfully discover 6 distinct human-interpretable strategies in the GRF 3-vs-1 scenario and 4 strategies in two 11-player GRF scenarios, namely counter-attack and corner, without any domain knowledge, which are substantially more than existing baselines.

## 2 RELATED WORK

Discovering diverse solutions has been a long-established problem (Miller & Shaw, 1996; Deb & Saha, 2010; Lee et al., 2022) with a wide range of applications in robotic control (Cully et al., 2015; Kumar et al., 2020), dialogues (Li et al., 2016), game AI (Vinyals et al., 2019; Lupu et al., 2021), design (Gupta et al., 2021) and emergent behaviors (Liu et al., 2019; Baker et al., 2020; Tang et al., 2021). Early works are primarily based on the setting of multi-objective optimization (Mouret & Clune, 2015; Pugh et al., 2016; Ma et al., 2020; Nilsson & Cully, 2021; Pierrot et al., 2022), which assumes a set of reward functions is given in advance. In RL, this is also related to reward shaping (Ng et al., 1999; Babes et al., 2008; Devlin & Kudenko, 2011; Tang et al., 2021). We consider learning diverse policies without any domain knowledge.

Population-based training (PBT) is the most popular framework for producing diverse solutions by jointly learning separate policies. Representative algorithms include evolutionary computation (Wang et al., 2019; Long et al., 2020; Parker-Holder et al., 2020b), league training (Vinyals et al., 2019; Jaderberg et al., 2019), computing Hessian matrix (Parker-Holder et al., 2020a) or constrained optimization with a diversity measure over the policy population (Lupu et al., 2021; Zhao et al., 2021; Li et al., 2021; Liu et al., 2021b). An improvement over PBT is to learn a latent variable policy instead of separate ones to improve sample efficiency. Prior works have incorporate different domain knowledge to design the latent code, such as action clustering (Wang et al., 2021), agent identities (Li et al., 2021) or prosocial level (Peysakhovich & Lerer, 2018; Baker et al., 2020). The latent variable can be also learned in an unsupervised fashion. DIYAN (Eysenbach et al., 2019) and its variants (Kumar et al., 2020; Osa et al., 2022) learns latent-conditioned policies by maxi-

mizing the mutual information between states and the latent variable. The discovered behaviors are primarily low-level motion skills rather than high-reward strategies (Eysenbach et al., 2022).

Iterative learning (IL) simplifies PBT by only optimizing a single policy subject to different diversity measures, such as maximum mean discrepancy (Masood & Doshi-Velez, 2019), Wasserstein distance on actions (Sun et al., 2020), and cross entropy (Zhou et al., 2022), which are often action-based. We adopt a purely state-based measure. Some other works require an expensive clustering process before each optimization iteration (Zhang et al., 2019) or domain-specific features (Zahavy et al., 2021) while we consider measures that can be efficiently optimized in an end-to-end fashion. Besides, Pacchiano et al. (2020) learns a kernel-based score function to guide policy optimization. The score function is conceptually similar to our Wasserstein-distance-based diversity measure but is applied to a parallel setting with more restricted expressiveness power.

## 3 PRELIMINARY

**Notation:** We consider Partially Observable Markov Decision Process (POMDP) (Spaan, 2012), defined by a tuple $\mathcal{M} = \langle \mathcal{S}, \mathcal{A}, \mathcal{O}, r, P, O, \nu, H \rangle$. $\mathcal{S}$ is the state space. $\mathcal{A}$ and $\mathcal{O}$ are the action and observation space. $r : \mathcal{S} \times \mathcal{A}^n \to \mathbb{R}$ is the reward function. $O : \mathcal{S} \to \mathcal{O}$ is the observation function. $H$ is the horizon. $P$ is the transition function. For state $s, s' \in \mathcal{S}$ and an action $a \in \mathcal{A}$, the transition probability from $s$ to $s'$ by executing action $a$ is $P(s' \mid s, a)$. At timestep $h$, the agent receives an observation $o_h = O(s_h)$ from the current state $s_h$ and outputs an action $a_h \in \mathcal{A}$ w.r.t. its policy $\pi : \mathcal{O} \to \triangle(\mathcal{A})$. The RL objective $J(\pi)$, i.e., expected return, is defined by $J(\pi) = \mathbb{E}_{(s_h, a_h) \sim (P, \pi)} \left[ \sum_{h=1}^{H} r(s_h, a_h) \right]$. The discounted factor is omitted here to simplify notations. The above formulation can be naturally extended to cooperative multi-agent settings, where $\pi$ and $R$ correspond to the joint policy and the shared reward. We follow the standard POMDP notations for conciseness and evaluate our algorithm in complex cooperative multi-agent scenarios since multi-agent games are substantially more challenging than single-agent ones.

Finally, in order to discover diverse strategies, we aim to learn a set of $M$ policies $\{\pi_i\}_{i=1}^{M}$ such that all of these policies are locally optimal under $J(\cdot)$ but mutually distinct subject to some diversity measure $D(\cdot, \cdot) : \triangle \times \triangle \to \mathbb{R}$, which captures the difference between two policies. We present two popular computation procedures for this purpose.

**Population-Based Training (PBT):** PBT is a straightforward formulation of the diversity discovery problem by jointly learning $M$ policies $\{\pi_i\}_{i=1}^{M}$ subject to pairwise diversity constraints, i.e.,

$$\max_{\pi_1, \ldots, \pi_M} \quad \sum_{i=1}^{M} J(\pi_i) \quad \text{s.t. } D(\pi_j, \pi_k) \geqslant \delta \quad \forall j, k \in [M], j \neq k, \tag{1}$$

where $\delta$ is a threshold. Despite a precise formulation, PBT poses severe optimization challenges.

**Iterative Learning (IL):** IL is a greedy approximation of PBT by iteratively learning novel policies. In the $i$-th ($1 \leqslant i \leqslant M$) iteration, IL solves the following constrained optimization problem

$$\pi_i^{\star} = \arg\max_{\pi_i} \quad J(\pi_i) \quad \text{s.t. } D(\pi_i, \pi_j^{\star}) \geqslant \delta \quad \forall 1 \leqslant j < i. \tag{2}$$

IL runs unconstrained RL at first and then solves incrementally more constrained problems.

**Action-Based Diversity Measure:** We briefly introduce the diversity measure in this paragraph. Many prior works define $D(\cdot, \cdot)$ over actions, which can be formally summarized by

$$D_{\mathcal{A}}(\pi_i, \pi_j) = \mathbb{E}_{s \sim q(s)} \left[ \tilde{D}_{\mathcal{A}} \left( \pi_i(\cdot \mid s) \| \pi_j(\cdot \mid s) \right) \right], \tag{3}$$

where $q : \triangle(\mathcal{S})$ denotes some specific state distribution, and $\tilde{D}_{\mathcal{A}}(\cdot \| \cdot) : \triangle \times \triangle \to \mathbb{R}$ measures the difference between action distributions. $\tilde{D}_{\mathcal{A}}$ can be any probability distance such as Wasserstein distance (Sun et al., 2020), Jensen-Shannon Divergence (Lupu et al., 2021), cross-entropy (Zhou et al., 2022), or simply the $L_2$ distance given a continuous action space (Parker-Holder et al., 2020b).

# 4 ANALYSIS OF EXISTING DIVERSITY-DISCOVERY APPROACHES

In this section, we conduct both theoretical and quantitative analysis of existing approaches to motivate our method. We first compare computation frameworks, namely PBT and IL, in Sec. 4.1 and then present concrete failure examples for action-based diversity measures in Sec. 4.2.

## 4.1 COMPUTATION FRAMEWORK: POPULATION-BASED OR ITERATIVE LEARNING?

**Theoretical Comparison:** We consider the simplest motivation example in the setting of linear programming to intuitively illustrate the computation challenges. We simply assume that $\pi_i$ is a scalar, and $J(\pi_i)$ is linear in $\pi_i$, and $D(\pi_i, \pi_j) = |\pi_i - \pi_j|$. In our definition, PBT involves $\Omega(M^2)$ variables in a single constrained optimization problem while IL involves $\Omega(M)$ variables in all. It is well-known that the complexity of linear programming is a high degree polynomial (degree 3 or higher depending on the algorithm) w.r.t. the number of variables (Bertsimas & Tsitsiklis, 1997). Therefore, even in the linear case, we can notice that more constraints can pose substantial challenges to the optimization problem. This issue can be more severe in RL due to complex solution space and large training variance.

Although IL can be optimized efficiently, it remains unclear whether IL, as a greedy approximation of PBT, can obtain solutions of comparable rewards. Fig. 1 shows the worst case in 1-D setting when the policies found by IL (green) can indeed have much lower rewards than the PBT solution (red) when subject to the same diversity constraint. However, we will show in the next theorem that IL is guaranteed to have no worse rewards than PBT by trading off half of the diversity.

**Theorem 4.1.** *Assume $D$ is a distance metric. Denote the optimal value of Eq.( 1) as $T_1$. Let $T_2 = \sum_{i=1}^{M} J(\tilde{\pi}_i)$ where*

$$\tilde{\pi}_i = \arg\max_{\pi_i} \quad J(\pi_i) \quad s.t. \quad D(\pi_i, \tilde{\pi}_j) \geqslant \delta/2 \quad \forall 1 \leqslant j < i \tag{4}$$

*for $i = 1, \ldots, M$, then $T_2 \geqslant T_1$.*

*Proof.* See Appendix E.1. □

The above theorem provides a quality guarantee for the IL solutions. The proof can be intuitively explained by the 1-D example in Fig. 1. Assuming the worst case where the first IL solution lies in the middle of a plateau with size $\delta$ (green 1), then the next solution with threshold $\delta$ must locate outside the plateau with a low reward. However, if the threshold is halved, the IL solutions are guaranteed to locate in the high-reward area (blue 1 and 2). Thm. 4.1 shows that, for any policy pool derived by PBT, we can always use IL to obtain another policy pool, which has *the same rewards* and *comparable diversity scores*. We remark that the worst case in Fig. 1 may not be common for RL environments in practice.

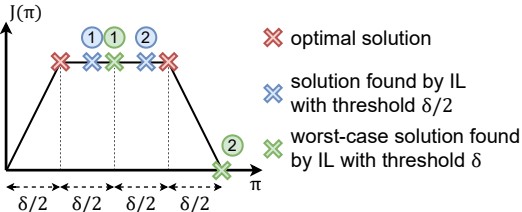

Figure 1: 1-D worst case of IL. With threshold $\delta$, IL finds solutions with inferior rewards. However, IL can find optimal solutions if the threshold is halved.

**Empirical Results:** We empirically compare PBT and IL in a 2-D navigation environment with one agent and $N_L$ landmarks (blue circles), as shown in Fig. 2. The reward is +1 if the agent successfully navigates to landmarks and 0 otherwise. Before training, landmark positions are randomly initialized subject to a pre-specified distance threshold per episode. We train $N_L$ policies using both PBT and IL to discover strategies towards each of these landmarks. Specifically, we simply take $D(\pi_i, \pi_j)$ as the $L_2$ distance of the final state reached by $\pi_i$ and $\pi_j$, i.e., $D(\pi_i, \pi_j) = \|s_H^{\pi_i} - s_H^{\pi_j}\|^2$. We solve this problem via Lagrangian multiplier with details in Appendix D.

Table 1: The number of discovered landmarks by PBT and IL across 6 seeds with standard deviation in the bracket.

| setting | PBT | IL |
|---|---|---|
| $N_L = 4$ | 2.0 (1.0) | **3.5**(0.5) |
| $N_L = 5$ | 2.2 (0.9) | **4.5**(0.5) |

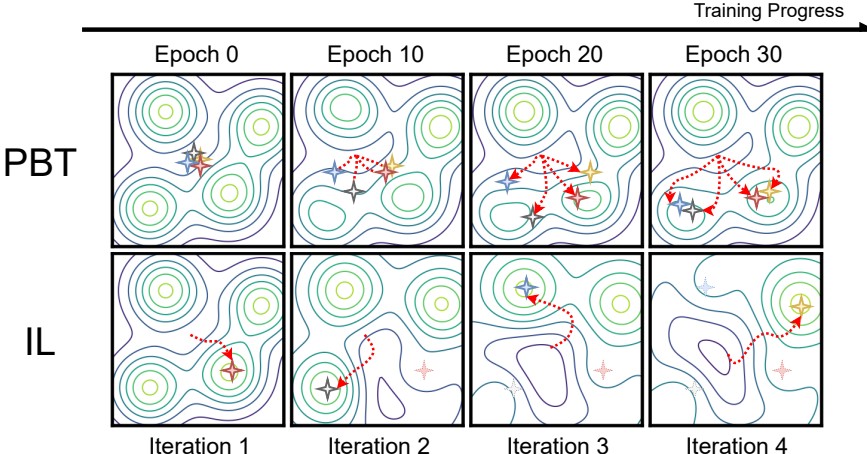

Figure 2: Illustration of the learning process of PBT and IL in a 2-D navigation environment with 4 modes. PBT will not uniformly converge to different landmarks as computation can be either too costly or unstable. By contrast, IL repeatedly excludes a particular landmark, such that policy in the next iteration can continuously explore until a novel landmark is discovered.

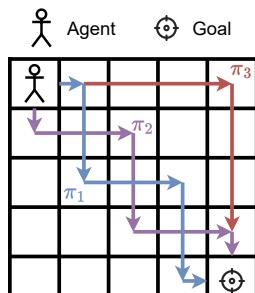

Table 2: Diversity measures of the grid-world example. Bold numbers indicate larger values. Computation details can be found in Appendix B. (KL=KL divergence, $JSD_\gamma$= generalized Jensen-Shannon Divergence, EMD=Earth Moving Distance)

| | human | action-based ($D_\mathcal{A}$) | | | | state-based ($D_\mathcal{S}$) | |
|---|---|---|---|---|---|---|---|
| | | KL | $JSD_1$ | $JSD_0$/EMD | $L_2$ norm | $L_2$ norm | EMD |
| $D(\pi_1, \pi_2)$ | small | $+\infty$ | $\mathbf{\log 2}$ | $\mathbf{1/2}$ | $\sqrt{7}$ | $2\sqrt{2}$ | 5.7 |
| $D(\pi_1, \pi_3)$ | large | $+\infty$ | $\mathbf{\log 2}$ | $1/8$ | 1 | $\mathbf{2\sqrt{6}}$ | $\mathbf{11.3}$ |

Figure 3: (left) A grid-world environment with size $N_G = 5$ and 3 different optimal policies. Intuitively, $D(\pi_1, \pi_2) < D(\pi_1, \pi_3)$ because $\pi_1$ (purple) and $\pi_2$ (blue) both move along the diagonal. However, action-based diversity measures can give $D_\mathcal{A}(\pi_1, \pi_2) \geqslant D_\mathcal{A}(\pi_1, \pi_3)$ (right), which motivates our proposal of state-distance based diversity measure.

Table 1 shows the number of discovered landmarks by PBT and IL. IL performs consistently better than PBT even in this simple example. We illustrate the learning process of PBT and IL in Fig. 2. IL, due to its computation efficiency, can afford to run longer iterations and tolerate larger exploration noises. Hence, it can converge easily to diverse solutions by imposing a large diversity constraint. The PBT, however, only converges when the exploration is faint, otherwise it diverges or converges too slowly.

## 4.2 Choice of Diversity Measure: Action-Based or State-Based?

We then analyze the impact of different diversity measures. We first show that action-based measures can often fail even for very simple tasks.

**Action-Based Measure:** Although action-based measures are easy to compute and widely used, we present concrete failure cases here. The first example is a single-agent grid-world with size $N_G$, where an agent spawns at the top left and needs to navigate to the bottom right. We consider three different policies shown in Fig. 3: $\pi_1$ (purple) and $\pi_2$ (blue) move along the diagonal while $\pi_3$ (red) moves along the boundary. Humans can naturally conclude that $\pi_3$ is visually different from $\pi_1$ and $\pi_2$, i.e., $D(\pi_1, \pi_2) < D(\pi_1, \pi_3)$, especially when $N_G$ is large. However, the actions of $\pi_1$ and $\pi_2$ along the trajectory are totally disjoint. Consequently, action-based measures will have a large value

on $D(\pi_1, \pi_2)$. We compute $D(\pi_1, \pi_2)$ and $D(\pi_1, \pi_3)$ based on popular action-based diversities measures in Table 2, where the obtained values largely violates human intuition.

Next, we consider a more realistic and complicated multi-agent football scenario in Fig. 4, where an idle player in the backyard takes an arbitrary action, such as "pass", "shoot" or "slide", without involving in the attack at all. Although the idle player stays still with no effect on the team strategy at all, action-based measures can produce high diversity scores when the idle player takes different duplicated actions, leading to visually indistinguishable solutions.

**State-Based Measure:** Based on the previous examples, we propose to focus on *states* rather than *action* when designing a diversity measure. Formally, denote the state distribution induced by $\pi$ as $\mu_\pi$. We define the *state-distance-based* diversity measure as

$$D_{\mathcal{S}}(\pi_i, \pi_j) = \mathbb{E}_{(s,s') \sim \gamma} \left[ g\left( d\left( s, s' \right) \right) \right]. \quad (5)$$

$d$ is a distance metric over $\mathcal{S} \times \mathcal{S}$. $g : \mathbb{R}^+ \to \mathbb{R}$ is a monotonic function. $\gamma \in \Gamma(\mu_{\pi_i}, \mu_{\pi_j})$ is a distribution over state pairs. $\Gamma(\mu_{\pi_i}, \mu_{\pi_j})$ denotes the collection of all distributions on $\mathcal{S} \times \mathcal{S}$ with marginals $\mu_{\pi_i}$ and $\mu_{\pi_j}$ on the first and second factors respectively.

Our proposed measure is solely defined over states and such a metric can impose a stronger and much smoother diversity constraint than existing action-based metrics. The state distance in the measure encourages the policies to reach visually different states leading to desired diversity. We compute two simple state-based measures, i.e., the $L_2$ norm and the Earth Moving Distance (EMD), for the grid-world example in Table 2, which is consistent with human intuition.

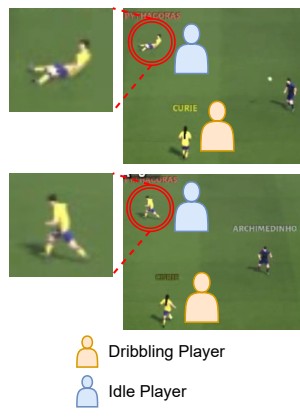

Dribbling Player

Idle Player

Figure 4: Duplicate actions in multi-agent football. For players who are not involved in the attack, actions like "pass", "shoot", and "slide" result in the same consequence. Diversity measures should not focus on these actions.

### 4.3 PRACTICAL REMARK

Based on the analysis in the above subsections, we conclude that PBT can pose severe optimization challenges, and that action-based diversity measures can often fail because they may not correctly reflect behavioral differences. By contrast, IL and state-based diversity measures are free from the above issues and should be preferred in challenging RL applications. Therefore, we consider how to develop a powerful algorithm for discovering diverse policies that can leverage *both* algorithmic design choices. In the next section, we combine these ideas with a theoretically sound optimization algorithm, Gradient Descent Ascent (GDA), towards an efficient and practical algorithm for learning diverse policies.

## 5 METHOD

### 5.1 ALGORITHM OVERVIEW

In this section, we develop a powerful diversity-driven RL algorithm, *State-based Intrinsic-reward Policy Optimization (SIPO)*, by combining IL and state-distance-based measures. SIPO runs $M$ iterations to discover $M$ distinct policies. At the $i$-th iteration, we solve Problem (2) by converting it into unconstrained optimization using the Lagrange method. The unconstrained optimization can be written as

$$\min_{\pi_i} \max_{\lambda_j \geq 0, 1 \leq j < i} -J(\pi_i) - \sum_{j=1}^{i-1} \lambda_j \left( D_{\mathcal{S}}(\pi_i, \pi_j^\star) - \delta \right) \quad (6)$$

where $\lambda_j$ $(1 \leq j < i)$ are Lagrange multipliers and $\{\pi_j^\star\}_{j=1}^{i-1}$ are previously obtained policies. We adopt two-timescale Gradient Descent Ascent (GDA) (Lin et al., 2020) to solve the above minimax optimization, i.e., performing gradient descent over $\pi_i$ and gradient ascent over $\lambda_j$ with different learning rates. We also clip the dual variables $\lambda$, which plays an important role both in our theorem

and in empirical convergence. However, $D_{\mathcal{S}}(\pi_i, \pi_j^{\star})$ cannot be directly optimized through gradient-based methods because it is related to the states visited by $\pi_i$. As a popular solution (Zhou et al., 2022), we cast $D_{\mathcal{S}}(\pi_i, \pi_j^{\star})$ as the summation of intrinsic rewards and optimize it via policy gradient. The pseudocode of SIPO can be found in Appendix G.

An important property of SIPO is the convergence guarantee. We present an informal illustration in Thm. 5.1 and present the formal theorem with proof in Appendix E.2.

**Theorem 5.1.** *(Informal) Under moderate assumptions, SIPO converges to a neighborhood of $\epsilon$-approximate KKT point.*

**Remark:** Please see the appendix for a detailed description of the assumptions and the proof. We assumed that the reward $J$ and the distance $D_{\mathcal{S}}$ are smooth in policies. In practice, this is true if the policy remains in a bounded region and the reward is continuous in state. The key step in the proof is to analyze the role of clipping the dual variables $\lambda$, which stabilizes the algorithm without hurting the optimality condition.

## 5.2 REALIZATION OF THE STATE-BASED MEASURE

Instead of directly defining $D_{\mathcal{S}}$, we define intrinsic rewards as illustrated in Sec. 5.1, such that $D_{\mathcal{S}}(\pi_i, \pi_j^{\star}) = \mathbb{E}_{s_h \sim \mu_{\pi_i}} \left[ \sum_{h=1}^{H} r_{\text{int}}(s_h; \pi_i, \pi_j^{\star}) \right]$. With this formulation, we can implement the following two types of diversity measures.

**RBF Kernel:** The most popular realization of Eq. (5) in machine learning is kernel functions. In this paper, we realize Eq. (5) as an RBF kernel on states. Formally, the intrinsic reward is defined by

$$r_{\text{int}}^{\text{RBF}}(s_h; \pi_i, \pi_j^{\star}) = \frac{1}{H} \mathbb{E}_{s' \sim \mu_{\pi_j^{\star}}} \left[ -\exp\left( -\frac{\|s_h - s'\|^2}{2\sigma^2} \right) \right] \tag{7}$$

where $\sigma$ is a hyperparameter controlling the variance.

**Wasserstein Distance:** For stronger discrimination power, we realize Eq. (5) as $L_2$-Wasserstein distance. According to the dual form (Villani, 2009), the intrinsic reward is defined by

$$r_{\text{int}}^{\text{WD}}(s_h; \pi_i, \pi_j^{\star}) = \frac{1}{H} \sup_{\|f\|_L \leqslant 1} f(s_h) - \mathbb{E}_{s' \sim \mu_{\pi_j^{\star}}} \left[ f(s') \right] \tag{8}$$

where $f : \mathcal{S} \rightarrow \mathbb{R}$ is a 1-Lipschitz function. Following Arjovsky et al. (2017), we implement $f$ as a neural network and clip parameters to $[-0.01, 0.01]$ to ensure the Lipschitz constraint. $r_{\text{int}}^{\text{WD}}$ utilizes a learnable scoring function $f$ and is more flexible in practice.

We name SIPO with $r_{\text{int}}^{\text{RBF}}$ and $r_{\text{int}}^{\text{WD}}$ ***SIPO-RBF*** and ***SIPO-WD*** respectively.

**Implementation** In the $i$-th iteration ($1 \leqslant i \leqslant M$), we learn an actor and a critic with $i$ separate value heads to accurately predict different return terms, including $i - 1$ intrinsic returns for the diversity constraints and the environment reward. The input of $r_{\text{int}}$ is the global state, which contains the state information of all the agents. To incorporate temporal information, we stack the recent 4 global states to compute intrinsic rewards and normalize the intrinsic rewards to stabilize training. In multi-agent environments, we learn an agent-ID-conditioned policy (Fu et al., 2022) and share the parameter across all agents. Our implementation is based on MAPPO (Yu et al., 2021) with more details in Appendix D.

## 6 EXPERIMENTS

We validate the effectiveness of SIPO in two complex multi-agent games: StarCarft Multi-Agent Challenge (SMAC) (Samvelyan et al., 2019) and Google Research Football (GRF) (Kurach et al., 2020). First, we show that SIPO can efficiently learn diverse strategies in all scenarios and outperform several baseline methods, including DIPG (Masood & Doshi-Velez, 2019), SMERL (Kumar et al., 2020), DvD (Parker-Holder et al., 2020b), and RSPO (Zhou et al., 2022). Then, we qualitatively demonstrate the emergent behaviors learned by SIPO, which are both *visually distinguishable*

Table 4: Number of visually distinct strategies in GRF discovered by different methods. Population size $M = 4$ in all cases. Details of the evaluation protocol can be found in Appendix B.

| | ours | | baselines | | | | random |
|---|---|---|---|---|---|---|---|
| | SIPO-RBF | SIPO-WD | DIPG | SMERL | DvD[1] | RSPO[2] | PG |
| *3v1* | **3.00(0.82)** | **3.00(0.00)** | 2.67(0.47) | 1.33(0.47) | **3.00(0.82)** | 2.33(0.47) | 2.67(0.47) |
| *CA* | **3.33(0.47)** | 3.00(0.82) | 2.33(0.47) | 0.75(0.43) | - | 2.67(0.47) | 1.67(0.47) |
| *corner* | 2.33(0.47) | **3.00(0.82)** | 2.33(0.47) | 0.75(0.43) | - | - | 2.33(0.47) |

[1] Training DvD in *CA* and *corner* requires >24GB GPU memory, which exceeds our memory limit.
[2] Not converged in *corner*.

and *human-interpretable*. Finally, we perform an ablation study over the building components of SIPO and show that both the diversity measure and GDA are critical to the performance.

All the algorithms including ablation variants run for the same number of environment frames on a desktop machine with a single NVIDIA RTX3090 GPU. All the quantitative results are repeated over 3 random seeds with standard deviation shown in brackets. Additional results in continuous control can be found in Appendix B.

## 6.1 COMPARISON WITH BASELINE METHODS

In SMAC, we only compare SIPO and RSPO, since RSPO outperforms other baselines (Zhou et al., 2022). We run both algorithms on an easy map, *2m_vs_1z*, and a hard map, *2c_vs_64zg*, across 4 iterations. Both algorithms can discover 4 distinct winning strategies. To perform further comparison, we compute the population diversity score based on $r_{\text{int}}^{\text{RBF}}$ (see definition in Appendix B).

The results in Table 3 show that SIPO can discover an even more diverse population than RSPO, even though RSPO explicitly forces all policies to output different actions. In GRF, we run all algorithms and train a population of $4$ in three academy scenarios, specifically "academy_3_vs_1_with_keeper" (*3v1*), "academy_counterattack_easy" (*CA*), and

Table 3: Population diversity in SMAC.

| map | RSPO | SIPO |
|---|---|---|
| *2m1z* | 0.035 (0.011) | **0.047**(0.011) |
| *2c64zg* | 0.292 (0.023) | **0.368**(0.013) |

"academy_corner" (*corner*). The GRF environment is more challenging than SMAC due to the large action space and the existence of duplicate actions. Table 4 compares the number of visually distinct policies discovered in the population. We present the population diversity scores in Appendix B. Our algorithm is the most efficient and robust — even in the challenging 11-vs-11 *corner* and *CA* scenario, SIPO can effectively discover different winning strategies in just a few iterations across different seeds. By contrast, baselines adopting action-based measures, e.g., DvD and RSPO, suffer from the issue of duplicate actions and tend to discover policies with slight distinctions. In addition, the mutual information objective in SMERL is sub-optimal (Eysenbach et al., 2022) and the MMD-based measure of DIPG may not impose a strong adversarial power on policies.

## 6.2 QUALITATIVE ANALYSIS

For SMAC, we present heatmaps of agent positions in Fig. 5. The heatmaps clearly show that SIPO can consistently learn novel winning strategies to conquer the enemy. Fig. 6 presents the learned behavior by SIPO in the GRF *3v1* scenario, where 3 attackers should collaborate to shoot under the defense of an opponent player and a goalkeeper. We can observe that agents have learned a wide spectrum of collaboration strategies across merely 7 iterations. Visualization results in *CA* and *corner* scenarios can be found in Appendix B.

Surprisingly, the strategies discovered by SIPO are both *diverse* and *human-interpretable*. We take the *3v1* scenario as an example. In the first iteration, all agents are involved in the attack such that they can distract the defender and obtain a high win rate. Similar strategies are discovered in the 4th and 5th iterations. The 2nd and the 6th iteration demonstrate an efficient pass-and-shoot strategy, where agents quickly elude the defender and score a goal. In the 3rd and the 7th iterations, agents learn smart "one-two" strategies to bypass the defender, which is a common tactic adopted by

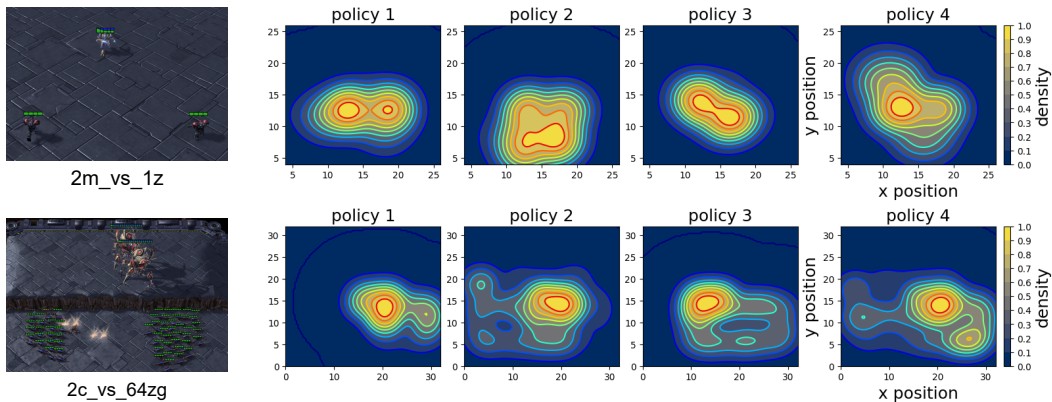

Figure 5: Heatmaps of agent positions in SMAC across 4 iterations with SIPO-RBF.

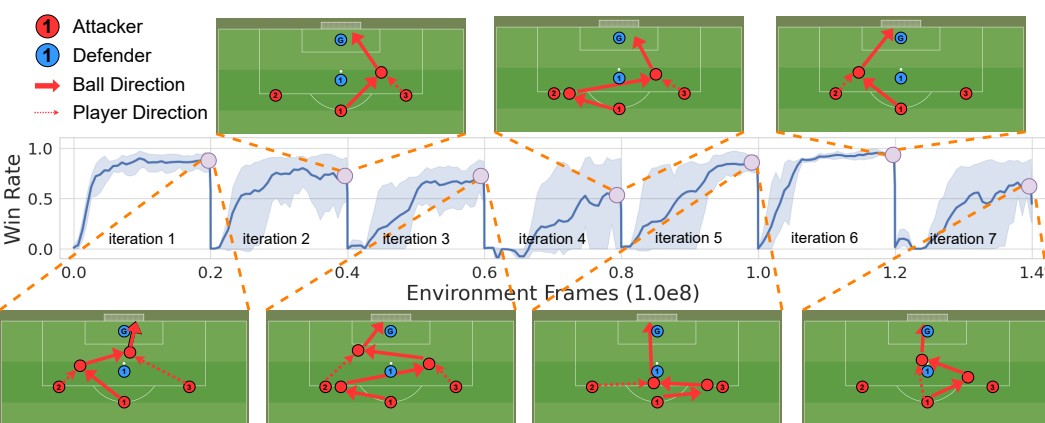

Figure 6: Evaluation winning rate and discovered strategies by SIPO-WD in the *3v1* scenario over 7 iterations. Each iteration runs for 20 million environment frames. The learning curve is averaged over 3 random seeds with standard deviation shaded. Strategies of seed 1 are shown.

professional human football players. To the best of our knowledge, SIPO may be the first algorithm that can discover such diverse human-like tactics in complex multi-agent RL environments.

## 6.3 ABLATION STUDY

We perform ablation studies by

- fixing the Lagrange multiplier (fix-L);
- replacing our proposed diversity measure with cross-entropy (CE);
- replacing GDA with the filtering-based method (filter);
- replacing IL with PBT (PBT).

Table 5: Number of visually distinct strategies in GRF discovered by ablations.

|  | ours | fix-L | CE | filter | PBT |
|---|---|---|---|---|---|
| *3v1* | **3.00(0.00)** | 1.00(0.00) | 3.00(0.82) | 1.33(0.47) | 2.67(0.47) |
| *CA* | **3.00(0.82)** | -[1] | 2.33(0.94) | 1.00(0.00) | -[2] |
| *corner* | **3.00(0.82)** | -[1] | 1.67(0.47) | 1.00(0.00) | -[2] |

[1] Not converged.
[2] Training requires > 24GB memory and exceeds our memory limit.

We apply these changes to SIPO-WD and report the number of visually distinct policies discovered by these methods in Table 5. Comparison between SIPO and CE demonstrates that the action-based cross-entropy measure may suffer from duplicate actions in GRF and produce nearly identical behavior by overly exploiting duplicate actions, especially in the *CA* and *corner* scenarios with 11 agents. Besides, the fixed Lagrange coefficient, the filtering-based method, and PBT are all detrimental to our algorithm. These methods also suffer from significant training instability. Overall, both the state-distance-based diversity measure and GDA are critical to the performance of SIPO.

## 7 CONCLUSION

In this paper, we tackle the problem of discovering diverse high-reward policy in complex RL scenarios. We present a thorough comparison between two popular computation frameworks for this problem, i.e., population-based training (PBT) and iterative learning, and show that, comparing with PBT, IL is much easy to optimize and can derive solutions with comparable quality to PBT. Moreover, we also demonstrate concrete failure cases for popular action-based diversity measure. Motivated by these insights, we combine IL with a diversity measure defined on state distance to develop *State-based Intrinsic-reward Policy Optimization (SIPO)*, which has provable convergence and can efficiently discover a wide spectrum of human-interpretable strategies in challenging multi-agent environments. We emphasize that the contribution of our work is much beyond the final algorithm SIPO. We believe our analysis on frameworks and diversity measure with concrete examples and theoretical justifications can bring useful insights to benefit the community for developing more powerful diversity-driven RL algorithms.

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

## A  PROJECT WEBSITE

Check https://sites.google.com/view/diversity-sipo for GIF demonstrations.

## B  ADDITIONAL RESULTS

### B.1  MORE QUALITATIVE RESULTS

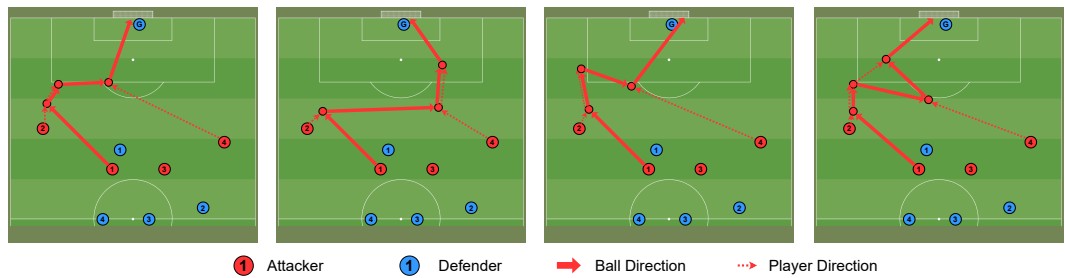

Figure 7: Visualization of learned behaviors in GRF *CA* across a single training trial.

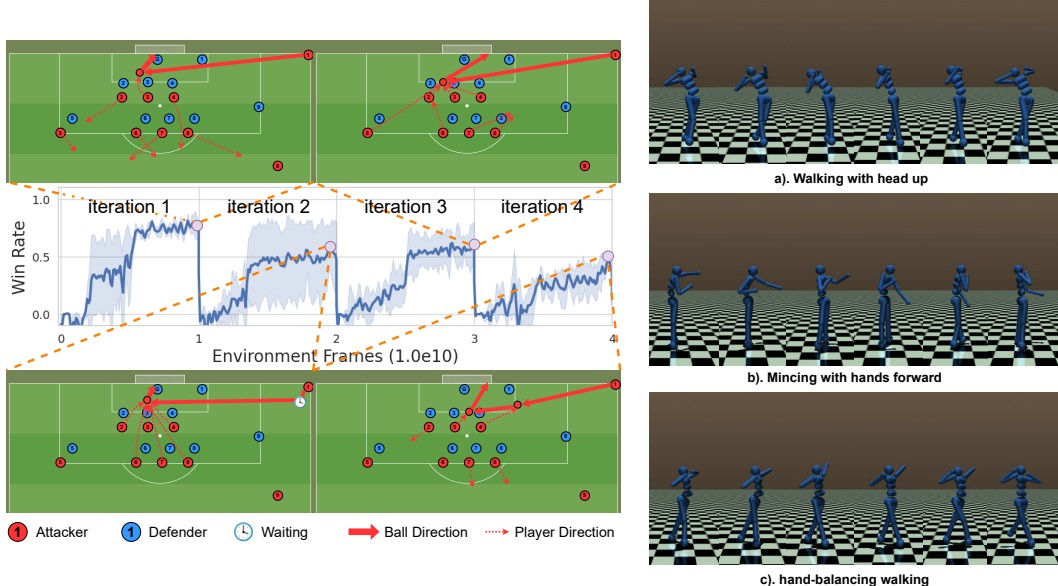

Figure 8: Visualization of learned behaviors in GRF *corner* (left) and MuJoCo Humanoid-v3 (right) across a single training trial.

We show visualization results in GRF *CA* and *corner* in Fig. 7 (left) and Fig. 8 (left). We also evaluate SIPO-WD in the most challenging continuous control environment, Humanoid-v3, across 3 iterations and visualize the learned behavior in Fig. 8 (right). SIPO-WD is able to produce diverse behaviors with different gaits. We additionally remark that the population diversity score is very close to 1 (such as 0.999) even when we repeatedly run PPO (Zhou et al., 2022). Hence, we do not report the population diversity score here.

### B.2  STATE-BASED POPULATION DIVERSITY

We define the pairwise difference between policies as

$$K_{ij} = K(\pi_i, \pi_j) = \mathbb{E}_{(s_h^1, s_2^h) \sim (P, \pi_i, \pi_j)} \left[ \exp\left( -\frac{\|s_h^1 - s_h^2\|^2}{2\hat{\sigma}^2} \right) \right]. \tag{9}$$

Table 6: Population diversity in GRF. Mean values averaged over 3 random seeds are shown with standard deviation in the brackets. Population size $M = 4$.

| | ours | | baselines | | | |
|---|---|---|---|---|---|---|
| | SIPO-RBF | SIPO-WD | DIPG | SMERL[1] | DvD[2] | RSPO[3] |
| *3v1* | 0.023(0.002) | 0.027(0.002) | 0.025(0.004) | 0.014(0.002) | 0.024(0.008) | 0.019(0.002) |
| *CA* | 0.972(0.022) | 0.945(0.050) | 0.919(0.004) | - | - | 0.899(0.058) |
| *Corner* | 0.213(0.088) | 0.278(0.086) | 0.154(0.021) | - | - | - |

[1] When conditioned on some specific latent variable, SMERL policy cannot even collect a single winning trajectory in *CA* and *Corner*. Therefore, we omit the result here.
[2] Training DvD in *CA* and *corner* requires >24GB GPU memory, which exceeds our memory limit.
[3] Not converged in *corner*.

Table 7: Population diversity and the number of distinct strategies in GRF *3v1* scenario with population size $M = 10$. Mean values averaged over 3 random seeds are shown with standard deviation in the brackets.

| | ours | | baselines | |
|---|---|---|---|---|
| | SIPO-RBF | SIPO-WD | DIPG | RSPO |
| population diversity | 0.291(0.053) | 0.399(0.040) | 0.251(0.018) | 0.272(0.036) |
| # strategies | 4.33(0.47) | 5.67(0.47) | 3.67(0.47) | 2.33(0.47) |

where $\hat{\sigma}$ is a scaling factor. Then, similar to Parker-Holder et al. (2020b), we compute the determinant of the matrix $K$ as the population diversity. $\hat{\sigma} = 1$ for Table 3, $\hat{\sigma} = 0.4$ for Table 6, and $\hat{\sigma} = 0.15$ for Table 7.

Similar to Table 3, we present the state-based population diversity score of GRF scenarios in Table 6. GRF scenarios are more challenging than SMAC and the trained policies may not always score a goal in each episode. (See evaluation winning rates in Table 8.) To reduce the variance, we collect 32 *winning* trajectories and compute population diversity scores on them.

### B.3 RESULTS WITH A LARGER POPULATION SIZE

To demonstrate the effectiveness of SIPO, we additionally conduct an experiment in the GRF *3v1* scenario with a population size $M = 10$. Baselines include DIPG and RSPO. We present the results in Table 7. Results show that SIPO clearly outperforms these baselines by consistently discovering one or more additional strategies.

Empirically, we find that there are 4 "primitive" strategies in the *3v1* scenario, which are pass-and-shoot (iteration 2 in Fig. 6), double-pass-and-shoot (iteration 1 in Fig. 6), and the corresponding mirror strategies. Across 10 iterations, baseline methods do not discover any strategies beyond these primitives, while SIPO is able to learn addition smart behaviors like "one-two" strategies (iteration 7 in Fig. 6).

### B.4 EVALUATION WIN RATE

The evaluation win rates of the demonstrated visualization results (Fig. 5, Fig. 6, Fig. 7, and Fig. 8) are shown in Table 8.

### B.5 COMPUTATION OF ACTION-BASED MEASURES IN THE GRID-WORLD EXAMPLE

We consider the policies illustrated in Fig. 9. These policies are all optimal since these actions only include "right" and "down" and actions on non-visited states can be arbitrary. We only mark actions on states visited by any of these 3 policies and actions on other states can be considered the same.

Table 8: Evaluation win rate (%) of the demonstrated visualization results. Averaged across 3 seeds with standard deviation shown in brackets.

| | SMAC | | GRF | | |
|---|---|---|---|---|---|
| | *2m1z* | *2c64zg* | *3v1* | *CA* | *corner* |
| $\pi_1$ | 100.0(0.0) | 98.1(2.1) | 92.3(6.2) | 48.2(10.4) | 78.2(16.2) |
| $\pi_2$ | 99.6(0.9) | 100.0(0.0) | 82.1(8.4) | 43.8(42.2) | 57.0(37.7) |
| $\pi_3$ | 100.0(0.0) | 96.9(3.3) | 90.7(1.1) | 54.7(30.6) | 55.7(20.8) |
| $\pi_4$ | 99.6(0.6) | 98.6(2.4) | 63.6(45.0) | 17.2(30.0) | 30.7(29.0) |
| $\pi_5$ | - | - | 85.4(9.1) | - | - |
| $\pi_6$ | - | - | 93.2(1.9) | - | - |
| $\pi_7$ | - | - | 64.6(32.5) | - | - |

### B.5.1 ACTION-DISTRIBUTION-BASED MEASURES

Action-distribution-based diversity measures can be defined as

$$D_{\mathcal{A}}(\pi_i, \pi_j) = \mathbb{E}_{s \sim q(s)} \left[ \tilde{D} \left( \pi_i(\cdot \mid s) \| \pi_j(\cdot \mid s) \right) \right], \tag{10}$$

where $\tilde{D}(\cdot, \cdot) : \triangle \times \triangle \to \mathbb{R}$ is a measure over action distributions and $q : \triangle(\mathcal{S})$ is a state proposal distribution. Here, we consider $q$ to be the joint state distribution visited by $\pi_i$ and $\pi_j$.

**KL Divergence**    KL divergence is defined by

$$D_{\text{KL}}\left( \pi_i(\cdot \mid s), \pi_j(\cdot \mid s) \right) = \int_{\mathcal{A}} \pi_i(a \mid s) \log \frac{\pi_i(a \mid s)}{\pi_j(a \mid s)} \mathrm{d}a.$$

When $\pi_j(a \mid s) = 0$ at any state $s$, KL divergence is $+\infty$. Since the trajectories of these policies have disjoint states, $D_{\mathcal{A}}^{\text{KL}}(\pi_1, \pi_2) = D_{\mathcal{A}}^{\text{KL}}(\pi_1, \pi_3) = +\infty$. Similar results can be obtained for cross-entropy.

**JSD$_\gamma$**    JSD$_\gamma$ was defined in Lupu et al. (2021) and we consider two special cases when $\gamma = 0$ and $\gamma = 1$.

As illustrated by Lupu et al. (2021), JSD$_0$ measures the expected number of times two policies will "disagree" by selecting different actions. On trajectories induced by $\pi_1$ and $\pi_2$, there are $4 + 4$ states that $\pi_1$ disagrees with $\pi_2$ ($\pi_1$ and $\pi_2$ are symmetric) and $D_{\mathcal{A}}^{\text{JSD}_0}(\pi_1, \pi_2) = 8/16 = 1/2$. Similarly, $\pi_1$ and $\pi_3$ only disagree at the initial state, therefore we have $D_{\mathcal{A}}^{\text{JSD}_0}(\pi_1, \pi_3) = 2/16 = 1/8$.

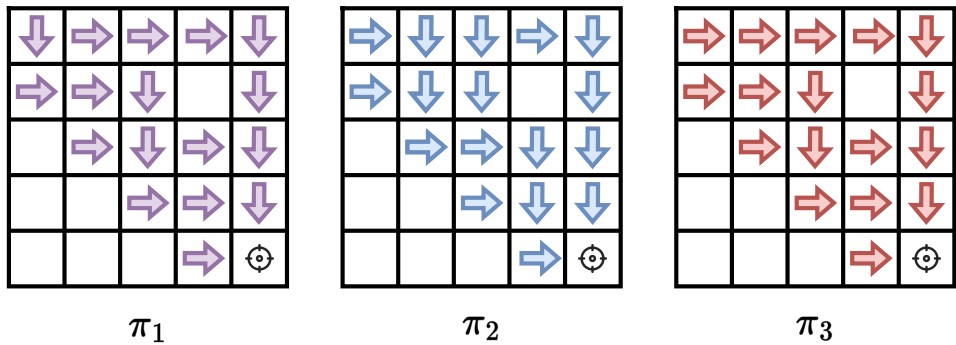

Figure 9: Policies in the grid-world example when $N_G = 5$.

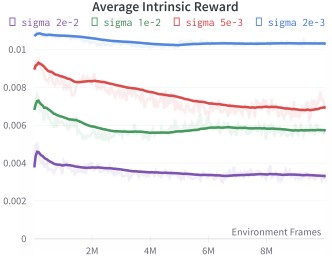

Figure 10: Average intrinsic reward during training $\pi_1$.

Table 9: The values of $\delta$ and $\alpha$ in different environments.

|  | football | | | smac | |
|---|---|---|---|---|---|
|  | *3v1* | *corner* | *CA* | 2m_vs_1z | 2c_vs_64zg |
| $\delta^{\text{WD}}$ | 0.004 | 0.01 | 0.012 | 0.02 | 0.2 |
| $\alpha^{\text{WD}}$ | 1 | 1 | 0.5 | 0.5 | 0.05 |
| $\delta^{\text{RBF}}$ | 0.03 | 0.01 | 0.015 | 0.002 | 0.001 |
| $\alpha^{\text{RBF}}$ | 0.001 | 0.001 | 0.001 | 0.001 | 0.001 |
| $\sigma^2$ | 0.02 | 0.02 | 0.02 | 0.02 | 0.02 |

$\text{JSD}_1$ is defined by

$$\text{JSD}_1(\pi_i, \pi_j) = -\frac{1}{2} \sum_{\tau_i} P(\tau_i \mid \pi_i) \sum_{t=1}^{T} \frac{1}{T} \log \frac{\pi_i(\tau_i) + \pi_j(\tau_i)}{2\pi_i(\tau_i)}$$

$$- \frac{1}{2} \sum_{\tau_j} P(\tau_j \mid \pi_j) \sum_{t=1}^{T} \frac{1}{T} \log \frac{\pi_i(\tau_j) + \pi_j(\tau_j)}{2\pi_j(\tau_j)}.$$

Since each of the policies considered only induces a single trajectory and $\pi_i(\tau_j) = 0$ $(i \neq j)$, we can easily compute

$$D_{\mathcal{A}}^{\text{JSD}_1}(\pi_1, \pi_2) = D_{\mathcal{A}}^{\text{JSD}_1}(\pi_1, \pi_3) = \log 2$$

**Wasserstein Distance** Wasserstein distance or Earth Moving Distance (EMD) is 1 if two policies disagree on a state and 0 otherwise. Therefore, it equals to $D_{\mathcal{A}}^{\text{JSD}_0}$.

### B.5.2 ACTION NORM

We embed the action "right" as vector $[1, 0]$ since it increases the x-coordinate by 1 and the action "down" as vector $[0, -1]$ since it decreases the y-coordinate by 1. This embedding can be naturally extended to a continuous action space with velocity action. Following Parker-Holder et al. (2020b), we compute the action norm over a uniform distribution on states. We can see that there are 7 states where $\pi_1$ and $\pi_2$ perform differently and 1 state (the initial state) where $\pi_1$ and $\pi_3$ perform differently. Therefore, we can get $D(\pi_1, \pi_2) = \sqrt{7}$ and $D(\pi_1, \pi_3) = 1$.

### B.5.3 STATE-DISTANCE-BASED MEASURES

**State $L_2$ Norm** Similar to action $L_2$ norm, we concatenate the coordinates instead of actions as the embedding and compute the $L_2$ norm between embedding.

**Wasserstein Distance** Wasserstein distance is tractable in the grid-world example. We consider 7 states (except the initial and final states) in each trajectory and compute the pair-wise distance as matrix $C_{14}$. Then we solve the following linear programming

$$\min_{\gamma} \quad \sum_{i,j} \gamma \odot C$$

$$\text{s.t.} \quad \gamma \mathbf{1}_{14} = a, \gamma^T \mathbf{1}_{14} = b$$

$$\gamma_{i,j} \geqslant 0, 1 \leqslant i, j \leqslant 14$$

where $\odot$ means element-wise multiplication, $\mathbf{1}_k$ is a $k$-dim all-one vector, $a_{14 \times 1} = [\mathbf{1}_k^T, \mathbf{0}_k^T]^T$ and $b_{14 \times 1} = [\mathbf{0}_k^T, \mathbf{1}_k^T]^T$ is the marginal state distribution of each policy.

### B.6 HOW TO ADJUST CONSTRAINT-RELATED HYPERPARAMETERS

Three hyperparameters are essential in the implementation of the intrinsic reward $r_{\text{int}}$: the threshold $\delta$, the intrinsic reward scale factor $\alpha$, and the variance factor $\sigma$ in $r_{\text{int}}^{\text{RBF}}$. These parameters differ

under different domains and must be adjusted individually. We find proper parameters by running two iterations without constraints and get two similar policies $\pi_0$ and $\pi_1$. We record $r_{\text{int}}$ during training $\pi_1$ and the trend is shown in Fig. 10. Not surprisingly, $r_{\text{int}}$ gradually decreases as training proceeds.

**Threshold** We set $\delta = c_1 D_{\mathcal{S}}(\pi_0, \pi_1)$. We try several different $c_1 \in \{1, 1.2, 1.4, 1.6, 1.8, 2.0\}$ and find that $c_1 = 1.2$ or $1.4$ are universal proper solutions for all the experimental environments.

**Intrinsic Scale Factor** We need to balance the intrinsic reward $r_{int}$ and the original reward $J$ so that neither of the two rewards can dominate the training process. Empirically, the maximums of the two rewards should be in the same order of magnitude. i.e., $\max_\pi J(\pi) = \alpha \times c_2 \lambda_{max} \delta$, where $c_2 = O(1)$. When $c_2$ is too large, the new-trained policy $\pi_j$ will oscillate near the boundary of $D(\pi_i, \pi_j) = \delta$ for some pre-trained policy $p_i$. Conversely, when $c_2$ is too small, the intrinsic reward $r_{int}$ cannot yield diverse strategies. In experiments, we set $c_2 = 0.8 \sim 1.0$.

**Variance Factor** We sweep the variance factor across $\{1e-3, 5e-3, 1e-2, 2e-2, 1e-3\}$ by training $\pi_1$ and observe the trend of intrinsic rewards. We find the steepest trend and select the corresponding $\sigma$. Empirically, we find that our algorithm performs robustly well when $\sigma^2 = 0.02$.

The $\delta$ and $\alpha$ of GRF and SMAC are listed in Table 9.

### B.7 Evaluation Protocol of Table 4 and Table 5

In *3v1* and *CA*, players perform passes and shoot in the front yard. We consider two strategies to be different if the resulting trajectories of ball movement are different, e.g. the ball is passed to different players or different players perform a shoot. In *Corner*, besides ball movement, we further categorize pass-and-shoot strategies according to the position of shooting in the penalty box (e.g., lower/middle/upper spot). All the authors perform independent evaluation based on this criterion and strong agreements are achieved. Please check our project website for GIF demonstrations.

## C Environment Details

### C.1 Details of the 2D Navigation Environment

The navigation environment has an agent circle with size $a$ and 4 landmark circles with size $b$. We pre-specify a threshold $c$ and constrain that the distance of final states reaching different landmarks must be larger than $c$. Correspondingly, landmark circles are randomly initialized by constraining the pairwise distance between centers to be larger than a threshold $c+2(a+b)$ such that the final-state constraint is valid. An episode ends if the agent touches any landmarks, i.e., the distance between the center of the agent and the center of the landmark $d < a+b$, or 1000 timesteps have elapsed. The observation space includes the positions of the agent and all landmarks, which is a 10-dimensional vector. The action space is a 2-dimensional vector, which is the agent velocity. The time interval is set to be $\Delta t = 0.1$, i.e., the next position is computed by $x_{t+1} = x_t + \Delta t \cdot v$. The reward is 0 if the agent touches the landmark and 0 otherwise.

### C.2 Details of SMAC, GRF, and MuJoCo

**SMAC** We adopt the SMAc environment in the MAPPO codebase[1] with the same configuration as Yu et al. (2021). The input of intrinsic rewards or diversity measure is the state of all allies, including positions, health, etc.

**GRF** We adopt the "simple115v2" representation as observation with both "scoring" and "checkpoint" reward. The reward is shared across all agents. The input of intrinsic rewards or diversity measure is the position and velocity of all attackers and the ball.

**MuJoCo** We use the Humanoid-v4 environment in OpenAI gym version 0.21.0 with the default configuration. To remove irrelevant or unchangeable features, we use the first 45-dimension of

---

[1]`https://github.com/marlbenchmark/on-policy`

Table 10: Hyperparameters in the 2D navigation environment.

| discount | GAE $\lambda$ | PPO epochs | clip parameter | entropy bonus | $\lambda_{max}$ | actor lr | critic lr | Lagrange lr | batch size |
|---|---|---|---|---|---|---|---|---|---|
| 0.997 | 0.95 | 10 | 0.2 | 0 | 10 | 3e-4 | 1e-3 | 0.5 | 4000 |

Table 11: Common hyperparameters for SIPO, baselines, and ablations.

| discount | GAE $\lambda$ | actor lr | critic lr | clip parameter | entropy bonus | GRF batch size | SMAC batch size |
|---|---|---|---|---|---|---|---|
| 0.99 | 0.95 | 5e-4 | 1e-3 | 0.2 | 0.01 | 9600 | 3200 |

observation as the input of intrinsic rewards and the semantic meaning can be found in `https://github.com/openai/gym/blob/master/gym/envs/mujoco/humanoid_v4.py`.

## D    IMPLEMENTATION DETAILS

### D.1    2D NAVIGATION

We apply PPO to optimize the policy and hyperparameters are summarized in Table 10. The applied algorithm is the same as SIPO (see Appendix G) except that the intrinsic reward is only computed at the last timestep.

### D.2    SIPO

We include all practical tricks mentioned in Yu et al. (2021) because we find them all critical to algorithm performance. We use separate actor and critic networks, both with hidden size 64 and a GRU layer with hidden size 64. The common hyperparameters for SIPO, baselines, and ablations are listed in Table 11. Other environment-specific parameters, such as PPO epochs and mini-batch size, are all the same as Yu et al. (2021). Besides, Table 9 and Table 12 lists some extra hyperparameters for SIPO.

Specific hyperparameters for baselines can be found in Appendix D.3.

### D.3    BASELINES

We re-implement all baselines with PPO based on the MAPPO (Yu et al., 2021) project. All algorithms run for the same number of environment frames.

**SMERL**    We implement SMERL (Kumar et al., 2020) with PPO, where the actor and the critic take as the input the concatenation of observation and a one-hot latent variable. The discriminator is a 2-layer feed-forward network with 64 hidden units. The learning rate of the discriminator is the same as the learning rate of the critic network. The input of the discriminator is the same as the input we use for SIPO-WD. The critic has 2 value heads for an accurate estimation of intrinsic return. Since SMERL trains a single latent-conditioned policy, we train SMERL for $M\times$ more environment steps, such that total environment frames are the same. The scaling factor of intrinsic rewards is $0.1$ and the threshold for diversification is $[0.81, 0.45, 0.72]$ ($0.9 \times [0.9, 0.5, 0.8]$) for "3v1", "counterattack", and "corner" respectively.

**TrajDi**    We also try TrajDi (Lupu et al., 2021) in the GRF domain. We sweep the action discount factor among $\{0.1, 0.5, 0.9\}$ and the coefficient of TrajDi loss among $\{0.1, 0.01, 0.001\}$. However,

Table 12: SIPO hyperparameters across all environments.

| $\lambda_{max}$ | Discriminator lr | Lagrangian lr |
|---|---|---|
| 1 | 1e-4 | 0.1 |

TrajDi fails to converge in the "3v1" scenario and exceeds the GPU memory in the "counterattack" and "corner" scenarios. Therefore, we exclude the performance of TrajDi in the main body.

**DvD**   We concatenate the one-hot actions along a trajectory as the behavioral embedding. The square of the variance factor, i.e., $\sigma^2$ in the RBF kernel, is set to be the length of behavioral embedding. We also use the same Bayesian bandits as proposed in (Parker-Holder et al., 2020b). Training DvD in "counterattack" and "corner" exceeds the GPU memory and we exclude the results in the main body.

**DIPG**   For DIPG (Masood & Doshi-Velez, 2019), we follow the opensource implementation[2]. We set the same variance factor in the RBF kernel as SIPO-RBF and apply the same state as the input of the RBF kernel. We sweep the coefficient of MMD loss among $\{0.1, 0.5, 0.9\}$ and find $0.1$ the most appropriate (larger value will cause training instability). We use the same method to save archived trajectories as SIPO and the input of the RBF kernel is the same as the input we use for SIPO-RBF. To improve training efficiency, we only back-propagate the MMD loss at the first PPO epoch, but the training is still much slower ($\sim$17h/iteration for *3v1*) than SIPO-RBF ($\sim$12 h/iteration for *3v1*).

**RSPO**   For RSPO (Zhou et al., 2022), we follow the opensource implementation[3] and use the same hyperparameters on the SMAC *2c_vs_64zg* map in the original paper for GRF experiments.

### D.4   Ablation Study Details

For the three ablation studies: fix-L, CE, and filter, we list the specific hyperparameters here:

- fix-L: we set the Lagrange multiplier to be $0.2$;
- CE: the threshold is $3.800$ and the intrinsic reward scale factor is $1/1000$ of that in the WD setting;
- filter: all the hyperparameters in the setting is the same as those in the WD setting.

## E   Proofs

### E.1   Proof of Theorem 4.1

**Theorem 4.1.**   *Assume $D$ is a distance metric. Denote the optimal value of Problem 1 as $T_1$. Let $T_2 = \sum_{i=1}^{M} J(\tilde{\pi}_i)$ where*

$$
\begin{aligned}
\tilde{\pi}_i = \arg\max_{\pi_i} \quad & J(\pi_i) \\
s.t. \quad & D(\pi_i, \tilde{\pi}_j) \geqslant \delta/2, \quad \forall 1 \leqslant j < i
\end{aligned}
\tag{3}
$$

*for $i = 1, \dots, M$, then $T_2 \geqslant T_1$.*

*Proof.* Suppose the optimal solution of Problem 1 is $\pi_1, \pi_2, ..., \pi_M$ satisfying $J(\pi_1) \geqslant J(\pi_2) \geqslant ... \geqslant J(\pi_M)$ and the optimal solution of Problem 4 is $\tilde{\pi}_1, \tilde{\pi}_2, ..., \tilde{\pi}_M$ satisfying $J(\tilde{\pi}_1) \geqslant J(\tilde{\pi}_2) \geqslant ... \geqslant J(\tilde{\pi}_M)$.

Assume the contrary that Thm. 4.1 is not true, which means $\sum_{i=1}^{M} J(\pi_i) = T_1 > T_2 = \sum_{i=1}^{M} J(\tilde{\pi}_i)$. Then we choose the smallest number $N \leqslant M$ that satisfies

$$
\sum_{i=1}^{N} J(\pi_i) > \sum_{i=1}^{N} J(\tilde{\pi}_i).
$$

By $T_1 > T_2$ we know that $N$ exists. In addition, because Problem 4 solves unconstrained RL in the first iteration, we know that $\tilde{\pi}_1 = \arg\max_{\pi} J(\pi)$ and then $J(\pi_1) \leqslant J(\tilde{\pi}_1)$. Therefore, $N \geqslant 2$.

---

[2]https://github.com/dtak/DIPG-public
[3]https://github.com/footoredo/rspo-iclr-2022

Suppose $J(\pi_N) \leqslant J(\tilde{\pi}_N)$. Then we have

$$\sum_{i=1}^{N-1} J(\pi_i) > \sum_{i=1}^{N-1} J(\tilde{\pi}_i).$$

Contradicting the fact that $N$ is the smallest number satisfies that equation.

Hence, we know that $J(\pi_N) > J(\tilde{\pi}_N)$. Then

$$J(\pi_1) \geqslant J(\pi_2) \geqslant ... \geqslant J(\pi_N) > J(\tilde{\pi}_N).$$

Consider the optimization problem of $\tilde{\pi}_N$:

$$\tilde{\pi}_N = \arg\max_\pi \quad J(\pi)$$
$$\text{s.t.} \quad D(\pi, \tilde{\pi}_j) \geqslant \delta/2, \quad \forall 1 \leqslant j < N.$$

This optimization does not find $\{\pi_1, \ldots, \pi_N\}$ but find $\tilde{\pi}_N$, which means that for each $\pi_i$, $1 \leqslant i \leqslant N$, there exists $1 \leqslant j_i < N$ such that $D(\pi_i, \tilde{\pi}_{j_i}) < \delta/2$. Otherwise, we will get the solution of the above problem as $\pi_i$ instead of $\tilde{\pi}_N$.

By the Pigeonhole Principle, we know that there exist two indexes $i_1 \in [N]$ and $i_2 \in [N]$ $(i_1 \neq i_2)$ such that $j_{i_1} = j_{i_2} = \hat{j}$. Then we have

$$D(\pi_{i_1}, \pi_{i_2}) \leqslant D(\pi_{i_1}, \tilde{\pi}_{\hat{j}}) + D(\pi_{i_2}, \tilde{\pi}_{\hat{j}}) < \delta/2 + \delta/2 = \delta,$$

where the inequality follows by the triangle inequality of the distance function.

It contradict with the fact that $D(\pi_{i_1}, \pi_{i_2}) \geqslant \delta$ in Problem 1.

Therefore, we prove the theorem $\sum_{i=1}^M J(\pi_i) = T_1 \leqslant T_2 = \sum_{i=1}^M J(\tilde{\pi}_i)$. $\qquad\square$

### E.2 PROOF OF THEOREM 5.1

In this section, we consider the $i$-th iteration of SIPO illustrated in Eq. (2). For the sake of simplicity, we use $a \leqslant \boldsymbol{\lambda} \leqslant b$ for vector $\boldsymbol{\lambda}$ to denote each component of $\boldsymbol{\lambda}$ satisfies $a \leqslant \lambda_i \leqslant b$, where $a, b \in \mathbb{R}$. We use $\pi$ to denote the policy we are optimizing, and $\pi_j$ $(1 \leqslant j < i)$ to denote a previously obtained policy. We denote the Lagrange function as $L(\pi, \boldsymbol{\lambda}) = -J(\pi) - \sum_{j=1}^{i-1} \lambda_j (D(\pi, \pi_j) - \delta)$.

To prove Theorem 5.1, we consider the following two optimization problems:

$$(\pi_i, \boldsymbol{\lambda}^\star) = \arg\min_\pi \max_{\boldsymbol{\lambda} \geqslant 0} L(\pi, \boldsymbol{\lambda}) \tag{11}$$

and

$$(\tilde{\pi}_i, \tilde{\boldsymbol{\lambda}}^\star) = \arg\min_\pi \max_{0 \leqslant \boldsymbol{\lambda} \leqslant \Lambda} L(\pi, \boldsymbol{\lambda}), \tag{12}$$

where $\Lambda = \frac{1}{\epsilon_0}$ and $\epsilon_0 > 0$ is sufficiently small.

**Assumption E.1.** $0 \leqslant J(\cdot) \leqslant 1$.

**Assumption E.2.** $\forall \boldsymbol{\lambda} \geqslant 0$, $L(\cdot, \boldsymbol{\lambda})$ is $l$-smooth and $\zeta$-Lipschitz.

**Lemma E.3.** $J(\pi_i) \leqslant J(\tilde{\pi}_i)$.

*Proof.* As the domain of $\boldsymbol{\lambda}$ in Eq. 12 is smaller than Eq. (11), we have $L(\pi_i, \boldsymbol{\lambda}) \geqslant L(\tilde{\pi}_i, \tilde{\boldsymbol{\lambda}})$.

By the fundamental property of Lagrange duality, we know that $L$ achieves its optimal value when $\boldsymbol{\lambda} = 0$ and the optimal value is $-J(\pi_i)$.

By the optimality of $(\tilde{\pi}_i, \tilde{\boldsymbol{\lambda}}^\star)$, we know that

$$-\sum_{j=1}^{i-1} \tilde{\lambda}_j^\star (D(\tilde{\pi}_i, \pi_j) - \delta) \geqslant 0. \tag{13}$$

Then we have

$$-J(\pi_i) = L(\pi_i, \boldsymbol{\lambda}^\star) \geqslant \tilde{L}(\tilde{\pi}_i, \tilde{\boldsymbol{\lambda}}^\star) = -J(\tilde{\pi}_i) - \sum_{j=1}^{i-1} \tilde{\lambda}_j^\star (D(\tilde{\pi}_i, \pi_j) - \delta) \geqslant -J(\tilde{\pi}_i).$$

$\square$

**Lemma E.4.** *Under Assumption E.1, $D(\tilde{\pi}_i, \pi_j) \geqslant \delta - \epsilon_0$, $\forall 1 \leqslant j < i$.*

*Proof.* We prove by contradiction.

Suppose there exists $1 \leqslant j_0 < i$, $D(\tilde{\pi}_i, \pi_{j_0}) < \delta - \epsilon_0$. Then we choose $\hat{\boldsymbol{\lambda}}$ such that

$$\hat{\lambda}_j = \begin{cases} \Lambda & j = j_0 \,, \\ 0 & 1 \leqslant j < i, j \neq j_0 \,. \end{cases}$$

By the Assumption E.1, Eq. (13), and $\Lambda = \frac{1}{\epsilon_0}$, we have

$$0 \geqslant -J(\pi_i) = L(\pi_i, \boldsymbol{\lambda}^\star) \geqslant L(\tilde{\pi}_i, \tilde{\boldsymbol{\lambda}}^\star) \geqslant L(\tilde{\pi}_i, \hat{\boldsymbol{\lambda}}) \geqslant -1 - \Lambda(D(\tilde{\pi}_i, \pi_{j_0}) - \delta) > 0.$$

That is a contradiction. So we have proved that

$$D(\tilde{\pi}_i, \pi_j) \geqslant \delta - \epsilon_0, \quad \forall 1 \leqslant j < i.$$

$\square$

**Lemma E.5.** *(Lin et al. (2020), Theorem 4.8) Under Assumption E.2, solving Eq. (12) via two-timescale GDA with learning rate $\eta_\pi = \Theta(\epsilon^4/l^3\zeta^2\Lambda^2)$ and $\eta_\lambda = \Theta(1/l)$ requires*

$$\mathcal{O}\left(\frac{l^3\zeta^2\Lambda^2 C_1}{\epsilon^6} + \frac{l^3\Lambda^2 C_2}{\epsilon^4}\right)$$

*iterations to converge to an $\epsilon$-stationary point $\pi_i^\star$, where $C_1$ and $C_2$ are the constants that depend on the distance between the initial point and the optimal point.*

**Theorem 5.1.** *Under moderate assumptions, SIPO converges to a neighbourhood of $\epsilon$-approximate KKT point.*

*Proof.* At the $i$-th ($1 \leqslant i \leqslant M$) iteration, SIPO solves the following constrained optimization

$$\min_{\pi_i} \quad -J(\pi_i)$$
$$\text{s.t.} \quad D(\pi_i, \pi_j) \geqslant \delta, \quad \forall 1 \leqslant j < i \,.$$

Consider the Lagrange function as $L(\pi, \boldsymbol{\lambda}) = -J(\pi) - \sum_{j=1}^{i-1} \lambda_j \left(D(\pi, \pi_j) - \delta\right)$. Denote the optimal solution of Eq. 11 and Eq. 12 as $(\pi_i, \lambda)$ and $(\tilde{\pi}_i, \tilde{\lambda})$ respectively.

By Lemma E.3 and Lemma E.4 we have

$$J(\pi_i) \leqslant J(\tilde{\pi}_i)$$
$$D(\tilde{\pi}_i, \pi_j) \geqslant \delta - \epsilon_0, \quad \forall 1 \leqslant j < i$$

and therefore we only need to consider the following nonconvex-concave optimization

$$\min_\pi \max_{0 \leqslant \boldsymbol{\lambda} \leqslant \Lambda} L(\pi, \boldsymbol{\lambda}) \,. \tag{14}$$

Following Lemma E.5, we know that the Two-Timescale GDA algorithm converges to an $\epsilon$-stationary point $\pi_i^0$. Denote $\Phi(\pi) = \max_{0 \leqslant \boldsymbol{\lambda} \leqslant \Lambda} L(\pi, \boldsymbol{\lambda})$ and $\|\cdot\|$ as the Euclidean distance. Using the property of $\epsilon$-stationary point $\pi_i^0$ in (Lin et al., 2020) (Lemma 3.8), we know that there exists $\hat{\pi}_i$ such that $\min_{\xi \in \partial \Phi(\hat{\pi}_i)} \|\xi\| \leqslant \epsilon$ and $\|\hat{\pi}_i - \pi_i^0\| \leqslant \epsilon/2l$.

From the definition of $L(\pi_i, \boldsymbol{\lambda})$, we know that $\hat{\pi}_i$ is an $\epsilon$-approximate KKT point of $J(\pi)$((Dutta et al., 2013)).

From the above deduction, the Two-Timescale GDA algorithm convergences to an $\epsilon$ neighbourhood of $\epsilon$-approximate KKT point of the above problem. The theorem then follows by applying the smoothness assumption.

$\square$

## F DISCUSSION

### F.1 THE FAILURE CASE OF STATE-BASED DIVERSITY MEASURES

A failure case of state-based diversity measures may be when the state space includes many *irrelevant features*. These features cannot reflect behavioral differences. If we run SIPO in such an environment, the learned strategies may be only diverse w.r.t these features and have little visual distinction. Like the famous noisy TV problem (Burda & Edwards, 2018), the issue of irrelevant features is intrinsically challenging for general RL applications, which cannot be resolved by using action-based diversity measures either.

Thanks to the advantages we discussed in the paper, we generally find that state-based metrics can be preferred in challenging RL tasks. Meanwhile, since the state dimension can be much higher than actions, it is possible that RL optimization over states may be accordingly more difficult than actions. In practice, we can design a feature selector for those most relevant features for visual diversity and run diversity learning over the filtered features. In SMAC and GRF, we utilize the agent features (excluding enemies) as the input of diversity constraint without further modifications, as discussed in Appendix D. We remark that even after filtering, the agent features remain high-dimensional while our algorithm still works well. Note that using a feature selector is a common practice in many existing domains, such as novelty search (Cully et al., 2015), exploration (Liu et al., 2021a), and curriculum learning (Campero et al., 2021). There are also works studying how to extract useful low-dimensional features from observations (Wu et al., 2019; Ghosh et al., 2019), which are orthogonal to our focus.

### F.2 THE DISTANCE METRIC IN STATE-BASED DIVERSITY MEASURES

In Sec. 5, we adopt the two most popular implementations in the machine learning literature, i.e., RBF kernel and Wasserstein distance, while it is totally fine to adopt alternative implementations. For example, we can learn state representations (e.g. auto-encoder, Laplacian, or successor feature) and utilize pair-wise distance or norms as a diversity measure. Similar topics have been extensively discussed in the exploration literature (Wu et al., 2019; Machado et al., 2020). We leave them as our future directions.

## G PSEUDOCODE OF SIPO

The pseudocode of SIPO is shown in Algorithm 1.

---

**Algorithm 1** SIPO (red for SIPO-RBF and blue for SIPO-WD)

---

**Input:** Number of Iterations $M$, Number of Training Steps within Each Iteration $T$.

**Hyperparameter:** Learning Rate $\eta_\pi$, Diversity Threshold $\delta$, Intrinsic Scale Factor $\alpha$, Lagrange Multiplier Upperbound $\lambda_{\max}$, Lagrange Learning rate $\eta_\lambda$, Wasserstein Critic Learning Rate $\eta_W$, RBF Kernel Variance $\sigma$.

1: Archived trajectories $X \leftarrow \varnothing$          ▷ to store states visited by previous policies
2: **for** iteration $i = 1, \ldots, M$ **do**
3:      Initialize policy $\pi_{\theta_i}$          ▷ initialization
4:      Initialize Wasserstein critic $f_{\phi_i}$
5:      **for** archive index $j = 1, \ldots, i - 1$ **do**
6:          Lagrange multiplier $\lambda_j \leftarrow 0$
7:      **end for**
8:      **for** Training step $t = 1, \ldots, T$ **do**
9:          Collect trajectory $\tau = \{(s_h, \boldsymbol{a}_h, r(s_h, \boldsymbol{a}_h))\}_{h=1}^H$
10:          **for** archive index $j = 1, \ldots, i - 1$ **do**
11:             $R_{\text{int}}^j \leftarrow 0$
12:          **end for**
13:          **for** timestep $h = 1, \ldots, H$ **do**          ▷ compute intrinsic reward
14:             $r_{\text{int},h} \leftarrow 0$
15:             **for** archive trajectory $\chi_j \in X$ **do**
16:                 $r_{\text{int},h}^j \leftarrow -\frac{1}{H|\chi_j|} \sum_{s' \in \chi_j} \exp\left(-\frac{\|s_h - s'\|^2}{2\sigma^2}\right)$
17:                 $r_{\text{int},h}^j \leftarrow \frac{1}{H}\left[f_{\phi_j}(s_h) - \frac{1}{|\chi_j|} \sum_{s' \in \chi_j} f_{\phi_j}(s')\right]$
18:                 $r_{\text{int},h} \leftarrow r_{\text{int},h} + \lambda_j \cdot r_{\text{int},h}^j$
19:                 $R_{\text{int}}^j \leftarrow R_{\text{int},h}^j + r_{\text{int},h}^j$
20:             **end for**
21:             $r_h \leftarrow r(s_h, \boldsymbol{a}_h) + \alpha \cdot r_{\text{int},h}$
22:          **end for**
23:          **for** archive index $j = 1, \ldots, i - 1$ **do**
24:             $\lambda_j \leftarrow \text{clip}\left(\lambda_j + \eta_\lambda\left(-R_{\text{int}}^j + \delta\right), 0, \lambda_{\max}\right)$      ▷ gradient ascent on $\lambda_j$
25:             $\phi_j \leftarrow \phi_j + \eta_W \frac{1}{H}\sum_{h=1}^H \nabla_{\phi_j}\left(f_{\phi_j}(s_h) - \frac{1}{|\chi_j|}\sum_{s' \in \chi_j} f_{\phi_j}(s')\right)$
26:             $\phi_j \leftarrow \text{clip}(\phi_j, -0.01, 0.01)$
27:          **end for**
28:          Update $\pi_{\theta_i}$ with $\{(s_h, \boldsymbol{a}_h, r_h)\}$ by PPO algorithm      ▷ policy gradient on $\theta_i$
29:      **end for**
30:      Collect many trajectories $\chi_i$          ▷ collect trajectories to approximate $d_{\pi_{\theta_i}}$
31:      $X \leftarrow X \cup \{\chi_i\}$          ▷ for the use of following iterations
32: **end for**

---

