# OpenReview forum: "Iteratively Learning Novel Strategies with Diversity Measured in State Distances"
_ICLR.cc/2023/Conference — Submitted to ICLR 2023_

### Official Review · Reviewer_GEJD · 2022-10-18

**Confidence:** 5
**Correctness:** 2
**Technical Novelty And Significance:** 2
**Empirical Novelty And Significance:** 2
**Recommendation:** 5

**Clarity, Quality, Novelty And Reproducibility:**

The ideas here are not novel, it is all taken from other recent papers and recombined. The analysis could be novel, but it is currently only a secondary contribution.

**Strength And Weaknesses:**

### Strengths

- This paper covers many very interesting areas, for example comparing state vs. action representations and parallel vs. sequential population learning. If these were executed in a more rigorous way it would make a great contribution.
- The results in Figure 6 are interesting.
- The experiments are in relevant and reasonably high dimensional tasks.
- Related work seems to be correctly cited.

### Weaknesses

- The biggest weakness here is that the paper is framed as a new algorithm to rule them all when it really doesn't need to be. The actual algorithm isn't particularly novel, it is very similar to approaches like DOMiNO (Zahavy et al), which train policies using state occupancy metrics. Instead, if the authors focused on analyzing 1. Parallel vs. Sequential population training and 2. State vs. action representations then this would be a good paper I would accept.
- I think it is wrong to say prior work "typically" use action based representations for diversity. This is really just DvD and TrajeDi. Many previous works, including the original Novelty search, DIAYN etc have used diversity in the state space/trajectories. It is not super novel to switch back to states.
- The failure mode of action based metrics presented in Figure 3 is not necessarily true. In DvD, the actions are compared for *the same states*, while here it is for different states. The only state where the policies take different actions here is s0, otherwise it is possible that both policies act the same way everywhere else. So concretely, I disagree with the L2 norm being 2\sqrt{2} for D(\pi_1, \pi_2) in the case where the states are sampled from a joint distribution of the state visitation from both policies.
- How does bolding work in the tables, e.g. Table 4 & 5? It seems like in both cases there are multiple cells with the same value but the one with your method is bolded and the others aren't. Seems totally unnecessary to try and look like you "win".

Minor issues (did not impact score):
- Section 3 title, "Preliminary" -> "Preliminaries"
- Sec 4.1 "motivation example" -> "motivating example"
- Sec 4.1 "worst case in 1-D setting" -> "in a 1-D setting"

**Summary Of The Paper:**

The paper explores methods for discovering diverse policies for RL problems, specifically focusing on MARL. This is well-motivated, as these types of methods have become increasingly prominent in recent years and played a part of large scale successes such as AlphaStar. The paper then disappointingly focuses on a new algorithm "SIP" which is rather similar to many other similar algorithms in the space. My suggestion would be to focus on the analysis of design choices and produce a useful paper for the community, rather than trying to win on some benchmarks which makes the results hard to trust and subsequently transfer.

**Summary Of The Review:**

Nothing in this paper is new and that is totally fine. But, what I dislike is the approach of claiming to invent a new algorithm that "wins" on a specific benchmark chosen by the authors and making the paper all about this algorithm. Instead, if the authors reframe this paper as an analysis of joint vs. sequential population training algorithms, and a comparison of state vs. action representations, with theory and experiments showing this trade-off then it would be a useful contribution and I'd likely vote to accept. The paper I just described would be interesting to many, more broadly useful, and might also get cited a fair amount.

---

> ### Author Response · Authors · 2022-11-14
> **Author Response to Reviewer GEJD (2/2)**
>
>
>
> > Many previous works, including the original Novelty search, DIAYN etc have used diversity in the state space/trajectories. It is not super novel to switch back to states.
>
> + We agree that the idea of using state-based metrics itself has been explored previously. However, we emphasize that our contributions are much beyond simply presenting a state-based metric:
> 1. We thoroughly analyze and point out that state-based measures can impose a stronger diversity constraint than action-based measures;
> 2. We suggest a distance measure between states be critical for diversity learning;
> 3. We prove that our algorithm can converge to optimal solutions under the state-based constraint (while DIAYN cannot [4]); and
> 4. We empirically validate that state-based measures can be effective in complex multi-agent games such as SMAC and GRF (where Novelty Search methods can fail to discover winning strategies).
>
> [4] Eysenbach, B., Salakhutdinov, R., & Levine, S. (2021, September). The Information Geometry of Unsupervised Reinforcement Learning. In International Conference on Learning Representations.
>
> > “So concretely, I disagree with the L2 norm being 2\sqrt{2} for D(\pi_1, \pi_2) in the case where the states are sampled from a joint distribution of the state visitation from both policies.”
>
> + We thank the reviewer for pointing out this issue. Regarding the sentence “the only state where the policies take different actions here is s0”, we remark this is not true. \pi_1 and \pi_3 only differ at the initial state while \pi_1 and \pi_2 differ at all diagonal states.
>
> + In the authors’ opinion, the state distribution for computing DvD score is a uniform distribution over all states, as in the original paper. In Fig. 9 in the appendix, we can see that there are 7 states where \pi_1 and \pi_2 perform differently and 1 state (the initial state) where \pi_1 and \pi_3 perform differently. Therefore, if we let the state distribution be uniform, we can get D(pi_1,\pi_2)=\sqrt{7} and D(pi_1,\pi_3)=1. We have updated our paper accordingly. This change does not influence our analysis.
>
> > How does bolding work in the tables, e.g. Table 4 & 5?”
>
> + We bolded the one with the largest mean and broke ties by bolding the one with the smallest variance. After reading the reviewer’s comment, we recognize that it can cause confusion to readers. Therefore, we decide to bold all numbers with the largest mean in the revised draft.

---

> ### Author Response · Authors · 2022-11-14
> **Author Response to Reviewer GEJD (2/2)**
>
> ### Thank you for your valuable feedback! We have updated our paper to address the concerns.
>
> > The biggest weakness here is that the paper is framed as a new algorithm to rule them all when it really doesn't need to be.
>
> > Instead, if the authors focused on analyzing 1. Parallel vs. Sequential population training and 2. State vs. action representations then this would be a good paper.
>
> + Thank you for the constructive comment. We fully agree that the analysis can be critical to the community and that’s why in Sec.4, we have performed analysis on 2-D navigation and grid-world examples to show algorithmic insights. We appreciate that the reviewer likes the analysis part of our paper.
>
> + However, we want to emphasize that, compared with traditional benchmark/analysis papers, most of which only discuss pros and cons, we have made a clear conclusion based on our analysis:  IL and state-based diversity measures should be preferred in challenging RL applications. That is the reason why we decided to move on to convert our conclusions to an algorithm, which is also novel in the literature to the best of our knowledge. Also, PBT/IL and state/action-based measures are two independent algorithmic components, which should be eventually unified in a practical algorithm. Therefore, we combine the better components (IL + state-based measure) and GDA to design a powerful diversity-driven algorithm. We have revised the paper to make the transition between sections more natural.
>
>
> + Beyond motivating examples, we have also conducted a corresponding analysis in GRF (Table 6). We additionally compare SIPO with the PBT alternative and present the results in the revised Table 6. SIPO performs better than the PBT/action-based alternative. Without a concrete algorithm, i.e., SIPO, we could not be able to conduct further analysis to validate our theoretical insights into really complex environments like GRF and SMAC.
>
> + In case of misunderstanding, we have slightly modified some transition paragraphs in the paper.
>
> > it is very similar to approaches like DOMiNO (Zahavy et al)
>
> + DOMiNO focuses on robustness and fast adaptation in locomotion tasks, so it adopts a reverse formulation, i.e., optimizing diversity with a hard constraint on episode return. Similar policies are acceptable in DOMiNO. By contrast, we focus on discovering diverse behaviors and fully accept sub-optimal strategies, leading to a hard constraint on diversity measures. In addition, DOMiNO lies in the PBT category while we iteratively explore the policy space. These formulation differences, lead to drastically different final solution sets.
>
> + Furthermore, DOMiNO follows [1], which requires the optimization objective to be convex and differentiable. Such an assumption does not apply to our diversity discovery setting since we optimize reward, which is neither convex nor differentiable, rather than diversity. Notably, our method can be directly applied to the setting considered by DOMiNO.
>
> [1] Zahavy, T., O'Donoghue, B., Desjardins, G., & Singh, S. (2021). Reward is enough for convex MDPs. Advances in Neural Information Processing Systems, 34, 25746-25759.
>
> > I think it is wrong to say prior work "typically" uses action-based representations for diversity. This is really just DvD and TrajeDi.
>
> + While there indeed exist other action-based works such as [2] and [3], we agree with the reviewer that the word ``typically’’ is not appropriate. We have revised our paper correspondingly. Thank you for your suggestion!
>
> [2] Sun, H., Peng, Z., Dai, B., Guo, J., Lin, D., & Zhou, B. (2020). Novel policy seeking with constrained optimization. arXiv preprint arXiv:2005.10696.
> [3] Zhou, Z., Fu, W., Zhang, B., & Wu, Y. (2021, September). Continuously Discovering Novel Strategies via Reward-Switching Policy Optimization. In International Conference on Learning Representations.

---

> ### Author Response · Authors · 2022-11-18
> **A Gentle Reminder from the Authors (2022/11/18)**
>
> Dear Reviewer GEJD:
>
> Since the deadline is approaching, we sincerely hope the reviewer can read our response and the revised paper. Please let us know if the reviewer has any comments about our response or any other additional concerns.
>
> Paper 3507 Authors

---

> > ### Comment · Reviewer_GEJD · 2022-11-19
> > **Not convinced**
> >
> > I appreciate the revisions but remain unconvinced. There are so many settings where state based diversity could be completely meaningless, for example in a procedurally generated environment every state will be novel. To say one method is strictly better is just plain wrong. Further, there is an additional benefit of PBT - the wall clock time is faster by a factor of the number of agents in the population. This is non trivial, in fact I would say in many settings the IL setup simply isn't even possible as it would take far too long to train. Net this paper seems to be trying to do something useful but provides a questionable conclusion in an important area.

---

> > > ### Author Response · Authors · 2022-11-19
> > > **We have updated our paper to clarify the confusion**
> > >
> > > We would like to thank the reviewer for your active involvement. **We have updated our paper < [in this link](https://mega.nz/file/4LEwhTbb#7qeAy-fo_6-xMW9LFZutE-nUvaXfe5JauYLMarizlso) > with revisions marked in purple**.
> > >
> > > We totally agree with the reviewer and apologize for the confusion. **We didn’t intend to claim that our algorithm is a universal solution to *ALL* RL applications.** Instead, we would like to provide **insightful suggestions for relevant practitioners under similar contexts**. We have revised our paper to make this clear. By the no-free-lunch theorem, there can always be certain cases that an algorithm is poor at. As a technical paper, we made our own algorithmic proposal for a collection of (very challenging) applications, i.e., games with object-centric state information (so not many irrelevant features). We remark that, although such settings do not require non-trivial feature extraction (compared with image-based MDPs in proc-gen settings), the games we considered are still particularly popular in RL literature and extremely complex in solution space. Note that even in these settings, our algorithm indeed achieves strong performances than alternative baselines (Table 1 & 5) as we discussed. So we believe our proposal can still contribute important insights to the community.
> > >
> > > Regarding the state-based metric, we agree that when an object-centric state representation isn’t available, representation learning can be necessary. Therefore, we additionally conduct an ablation study regarding state input of the diversity measure in Appendix B.8, showing that our algorithm can still work effectively with a **learned feature** (the Wasserstein discriminator network). Note that learning effective features in visual RL remains an active research domain and our algorithm can be naturally combined with those advances. Again, we would like to remark that tackling high-dimensional state is a fundamental challenge for RL, which cannot be solved by action-based metrics solely. We have clarified the problem setting in Sec. 3 and put more discussions in the conclusion section.
> > >
> > > Regarding the benefits of PBT, we agree with the reviewer that when the computation budget is sufficient (e.g., many works done by Google and DeepMind), PBT can be always preferred. Again, we clarify that we didn’t intend to claim that IL is universally better. Instead, as we discussed in the paper, PBT may incur substantial optimization difficulties (Sec 4.1) for RL. Empirically, a single RL run of the policy pool may often lose diversity (Fig 2). Hence, although IL requires sequential training, it discovers a substantially more diverse solution set in terms of sample complexity. This is the same trade-off between on-policy RL algorithms (PPO) and off-policy methods (SAC). PPO can be extremely powerful if the batch size is sufficiently large (like AlphaStar and OpenAI Five) while SAC is still preferred often in an academic setting for its sample efficiency.
> > >
> > >  We have put discussions on this in the conclusion section as well.

---

> > > > ### Comment · Reviewer_GEJD · 2022-11-25
> > > > **Yet more unsubstantiated arguments, missing the point**
> > > >
> > > > First, this is plain wrong. The generations in XLand were explicitly distilling from the previous, if instead there is a diversity loss it would do the exact opposite and make it more like training from scratch which would defeat the point. To get diversity XLand used PBT. It is probably a good idea to be sure about things before throwing them out as an argument just to get your paper accepted!
> > > >
> > > > Second, you say you showed that there is some representation learning, but I explicitly said this method doesn't scale to procedurally generated environments, and you don't have any of those in your paper. Sure, if it is possible to learn meaningful representations of the states to then use it for diversity then that might work, but that is not what your paper does at all because you operate in visually simple domains.

---

> > > ### Author Response · Authors · 2022-11-25
> > > **Further clarification (updated in italic)**
> > >
> > > Dear Reviewer GEJD,
> > >
> > > We thank you again for joining the discussion. Please allow us to add two points if our previous response didn't fully address your concerns.
> > >
> > > > Further, there is an additional benefit of PBT - the wall clock time is faster by a factor of the number of agents in the population.
> > >
> > > + We agree with the reviewer that PBT allows a natural parallelization implementation. However, this parallelization does not come for free, because each agent needs to talk with every other agent in order to impose the diversity constraint. This will not only significantly increase the communication cost, and also incurs additional number of iterations (propotional to n, see for example Lemma 5 [1]) to convergence due to additional constraints.  Consequently, even with additional computing nodes, PBT is unlikely to improve the _run time toward convergence_ significantly as the convergence will be slow due to additional constraints.
> > >
> > > > in a procedurally generated environment every state will be novel
> > >
> > > + We apologize if our message were not clearly delivered. In our algorithm, the novelty of a state is not only measured by occupancy (i.e. whether the state is seen), but also by how much it differs from the previously seen states (i.e. the distance in state space). This design is reflected in our choice of probability distance--we used Wasserstein distance instead of KL or TV distance. Hence we say in title that our algorithm measure novelty in state "distances".
> > >
> > > We are happy to address any further concerns.
> > >
> > >
> > >
> > >
> > > [1]Nedic, Angelia, and Asuman Ozdaglar. "Distributed subgradient methods for multi-agent optimization." IEEE Transactions on Automatic Control 54.1 (2009): 48-61.

---

> > > > ### Comment · Reviewer_GEJD · 2022-11-25
> > > > **These answers don't mean anything!**
> > > >
> > > > 1) PBT "is unlikely to improve the runtime" -> totally unsubstantiated, you have zero experiments to show this. It is a huge weakness of iterative training vs. PBT, which is why PBT is used in many large scale projects and iterative training is not. This makes many of the findings in this paper largely irrelevant for anyone actually running these methods at scale.
> > > >
> > > > 2) Using a slightly fancier distance metric doesn't change anything about the fact you may have very different observations with the same semantic meaning. Your answer misses the point of the concern I had about PCG environments.

---

> > > > > ### Author Response · Authors · 2022-11-25
> > > > > **We hope the reviewer could first check our previous poist next to this thread.**
> > > > >
> > > > > Dear Reviewer GEJD,
> > > > >
> > > > > We sincerely hope that you could first check our [previous post](https://openreview.net/forum?id=OfaJyiYonBk&noteId=gJNIIEvcMnP), which is the one underneath this thread. Most of the main concerns have been answered previously. The bullet points in this thread are additional comments.
> > > > >
> > > > > Url to our previous post: https://openreview.net/forum?id=OfaJyiYonBk&noteId=gJNIIEvcMnP
> > > > >
> > > > > A few further clarifications on this thread:
> > > > >
> > > > > Regarding that _PBT unlikely to be improved_: We are specifically talking about the **convergence iterations with mathematical proofs (e.g. in [1])**. In practice, we do agree that PBT can save wall-clock time, as discussed in the previous thread,  if (1) sufficient compute is available, and (2) we stop early before (theoretical) convergence.
> > > > >
> > > > > We also would like to point out that **iterative policy learning has been recently adopted in some large-scale projects by DeepMind**, such as generational training [2] and DeepNash [3], to persue better solution quality.
> > > > >
> > > > > Regarding the distance measure, we respectively disagree with the judgement that this merely a fancy metric. Wasserstain distance is a **learning-based** metric which implicitly **learns** a low-dimensional representation. This is our attempt to incorporate representation learning into our framework.
> > > > >
> > > > > [1] Nedic, Angelia, and Asuman Ozdaglar. "Distributed subgradient methods for multi-agent optimization." IEEE Transactions on Automatic Control 54.1 (2009): 48-61.
> > > > >
> > > > > [2] Team, O. E. L., Stooke, A., Mahajan, A., Barros, C., Deck, C., Bauer, J., ... & Czarnecki, W. M. (2021). Open-ended learning leads to generally capable agents. arXiv preprint arXiv:2107.12808.
> > > > >
> > > > > [3] Perolat, J., de Vylder, B., Hennes, D., Tarassov, E., Strub, F., de Boer, V., ... & Tuyls, K. (2022). Mastering the Game of Stratego with Model-Free Multiagent Reinforcement Learning. arXiv preprint arXiv:2206.15378.

---

### Official Review · Reviewer_aXw1 · 2022-10-25

**Confidence:** 4
**Correctness:** 3
**Technical Novelty And Significance:** 3
**Empirical Novelty And Significance:** 3
**Recommendation:** 6

**Clarity, Quality, Novelty And Reproducibility:**

Clarity: The author has a very clear display of the structure of SIPO,  and the graph is given from a geometrical point of view, which is very easy to understand, and the description of the experimental settings and details is also relatively clear.
Quality: The charts are well made, and the visual effects of the experimental environment constructed are also very good.
Novelty: SIPO solves the problem of discovering diverse high-reward policies in complex RL scenarios of PBT and gives an explanation from a novel geometrical perspective.
Reproducibility: The author provided the original data, graphs, and code, the reproducibility is worthy of recognition.


**Strength And Weaknesses:**

a)	Strength:
This paper addresses a very important problem in reinforcement learning: policies with similar rewards may have fundamentally different behaviors. Therefore, the authors propose SIPO to discover as many different policies as possible. This paper objectively expounds on the disadvantages of the PBT method and proposes a very novel method that mathematically gives a state-based diversity measure and proves the convergence. Through experimental results, they find that SIPO is able to consistently derive strategically diverse and human-interpretable policies that cannot be discovered by existing baselines. The graphs are very clear. Figure 2 compares the training process of SIPO and PBT from a geometric perspective, which proves the advantages of SIPO and gives a lot of inspiration. The RL environments are also complex, which can fully prove the ability of the SIPO, and provides a well-made demo.

b)	Weaknesses:
When measuring Action-Based diversity, the author claims that A can be any distance metric. For different distance metrics, theoretical analysis is no problem, but will it affect the final experimental results? Maybe the author can consider adding these Experiments and Demonstrations. There are also many improvements to the PBT method in the future, perhaps the author can consider making further comparisons to reflect the advantages of SIPO.


**Summary Of The Paper:**

Combining iterative learning (IL) and the state-based diversity measure, this paper proposed a new framework called a powerful diversity-driven RL algorithm, State-based Intrinsic-reward Policy Optimization (SIPO), with provable convergence properties. IL repeatedly learns a single novel policy that is sufficiently different from previous ones. The paper proved that, for any policy pool derived by PBT, they can always use IL to obtain another policy pool of the same rewards and competitive diversity scores. In addition, they also present a novel state-based diversity measure with two tractable realizations, which can impose a stronger and much smoother diversity constraint than existing action-based metrics. It has been verified experimentally that SIPO is able to consistently derive strategically diverse and human-interpretable policies that cannot be discovered by existing baselines.

**Summary Of The Review:**

This paper addresses the problem of discovering diverse high-reward policies in complex RL scenarios. An in-depth comparison of PBT and IL is carried out, and it is found that IL is easier to optimize and can obtain comparable solutions to PBT. Furthermore, the authors illustrate the necessity of combining IL with a diversity measure defined on the state distance through specific failure cases of action-based diversity measures, and finally propose a state-based intrinsic reward policy optimization algorithm (SIPO) and prove that The proposed framework and the analysis of the diversity measure will bring a lot of inspiration to future work on policy diversity and PBT.

---

> ### Author Response · Authors · 2022-11-14
> **Author Response to Reviewer aXw1**
>
> ### Thank you for your appreciation of our work. Please see our detailed explanations below.
>
> > For different distance metrics, theoretical analysis is no problem, but will it affect the final experimental results?
>
> + In the paper, we adopt the two most popular implementations in the machine learning literature, while it is totally fine to adopt alternative implementations. For example, we can learn state representations (e.g. auto-encoder, Laplacian, or successor feature) and utilize pair-wise distance or norms as a diversity measure. Similar topics have been extensively discussed in the exploration literature [1,2]. We leave them as our future directions.
>
> [1] Machado, M. C., Bellemare, M. G., & Bowling, M. (2020, April). Count-based exploration with the successor representation. In Proceedings of the AAAI Conference on Artificial Intelligence (Vol. 34, No. 04, pp. 5125-5133).
>
> [2] Wu, Y., Tucker, G., & Nachum, O. (2018, September). The Laplacian in RL: Learning Representations with Efficient Approximations. In International Conference on Learning Representations.
>
> > There are also many improvements to the PBT method in the future, perhaps the author can consider making further comparisons to reflect the advantages of SIPO.
>
> + We would like to thank you for the constructive comment. Under our setting (discovering novel policies with RL), PBT methods are usually prohibitively expensive when the population size M is large, and we believe IL is clearly advantageous over PBT (e.g. we can open-endedly train many policies without reaching the resource limit). However, we agree with the reviewer that PBT can be preferred under certain scenarios. We also replace the computation framework of SIPO with PBT and perform an additional ablation study in the GRF 3v1 scenario. Agents trained by PBT methods can sometimes fail to learn a winning strategy and the number of distinct strategies is lower than the original SIPO, which implies the optimization challenge posed by PBT. The above results have been added to Table 6.

---

> > ### Comment · Reviewer_aXw1 · 2022-11-25
> > **Response to authors**
> >
> > After reading the author's response and other responses in detail, I decided to update my score.

---

### Official Review · Reviewer_nUUy · 2022-11-02

**Confidence:** 3
**Correctness:** 3
**Technical Novelty And Significance:** 3
**Empirical Novelty And Significance:** 2
**Recommendation:** 6

**Clarity, Quality, Novelty And Reproducibility:**

- The paper is not hard to follow and presentation is quite good.
- Even though I did not check all proofs in detail, the theory seems to be sound. One issue is that the papers make some strong statements that are not necessarily backed by significant evidence (see particularly first and second weaknesses above). I think more precise statements would strengthen the paper.
- The empirical analysis shows that SIPO can indeed learn diverse policies; however, I am not convinced that it provides strong evidence for the consistent superiority of SIPO compared to RSPO.
- The work is quite novel, in the sense that it provides several interesting insights about IL (vs. PBT), and additionally it is the first work (to the best of my knowledge) to rigorously analyze the convergence properties of IL with GDA.
- The authors provide details on experimental settings and hyperparameters to facilitate reproducibility.

**Strength And Weaknesses:**

Strengths
- The work provides several interesting insights: (i) the fact that IL can in fact achieve rewards similar to PBT by just trading-off half the diversity; (ii) the fact that IL can converge easily to diverse solutions by imposing a large diversity constraint, as opposed to PBT which suffers from slow convergence (or even divergence); (iii) the fact that in certain scenarios action-based diversity may produce similar policies, which motivates the authors to introduce a state-based diversity measure.
- The connection of SIP with intrinsic rewards is interesting. Intrinsic rewards and curiosity are often used to signal whether the agent has encountered a known or surprising state, so they fit well with the state-based diversity.
- SIPO comes with theoretical guarantees (when coupled with GDA).
- The two multi-agent games considered in the experimental evaluation are quite complex, and SIPO is able to achieve good results.

Weaknesses
- I understand that in the scenarios considered in this work, a state-based diversity measure is more appropriate. However, is this always the case? For example, one case where a state-based diversity measure would not make sense is when there are many irrelevant states. In that case, even if a new policy is quite diverse in terms of the irrelevant states, it may not be much more interesting than the existing policy. Do the authors make the claim that state-based diversity should always be preferred to action-based diversity? That would be a strong statement, and I am not sure enough evidence is provided in the test for such a claim.
- The authors claim that the strategies discovered by SIPO are not only diverse but human-interpretable as well. I do not dispute that this is the case in the studied scenarios. But I do not see how the proposed framework can generally guarantee interpretability. Is it sufficient to just enforce state-based diversity for this purpose? I do agree that SIPO would be expected to result in diverse policies, but I personally fail to see how human-interpretability can arise out of SIPO.
- In Table 4, the authors mention the number of visually distinct strategies discovered by different methods. Maybe I missed that in the text, but how did the authors come up with the exact number? This seems a very subjective metric, since different observers may report a different number. I was wondering whether some other methodology was followed instead.
- When it comes to comparing SIPO with other competitors (including RSPO), I am not sure that the current empirical evaluation does a very convincing job. Tables 3 and 4 indeed show improved numbers for SIPO, but RSPO also performs quite well (e.g., in Table 4). The empirical analysis shows to some extent that SIPO produces diverse policies, but I am not so sure that it shows very strongly that SIPO is consistently better than RSPO.

**Summary Of The Paper:**

An important problem in RL is to compute a diverse set of strategies that optimize the (total) reward. One strategy is to simultaneously optimize over all strategies, as with population-based training (PBT), but this can be computationally challenging. An alternative is to use a greedy algorithm, as with iterative learning (IL), which iteratively learns a new policy that is sufficiently diverse from all policies computed so far. The paper sheds light on the performance guarantees and the convergence properties of IL methods. Furthermore, it argues that, at least in the scenarios considered in the paper, a diversity measure based on state distances is more meaningful than action-based diversity. The authors combine IL and state-based diversity measures, and propose a new framework, State-based Intrinsic-reward Policy Optimization (SIPO), for discovering diverse RL strategies in an iterative fashion. For optimization, they use two-timescale gradient descent ascent (GDA), which comes with convergence guarantees. SIPO is empirically evaluated on two challenging multi-agent environments, namely, Star-Craft Multi-Agent Challenge and Google Research Football, where it is able to discover diverse and human-interpretable strategies.


**Summary Of The Review:**

Overall, I tend to be positive about the work because it provides various a number of interesting insights together with a rigorous analysis of an IL algorithm (SIPO with GDA). The results seem to confirm that SIPO can compute diverse policies in the multi-agent games of the evaluation. That said, the paper makes various strong claims, which may not be fully substantiated in the text. It would make sense to revisit some of these statements to increase precision. Furthermore, even though SIPO can get slightly better numbers than RSPO in the evaluation, I have the feeling that the empirical analysis does not provide very strong evidence for the superior performance of SIPO compared to RSPO.

---

> ### Author Response · Authors · 2022-11-14
> **Author Response to Reviewer nUUy (2/2)**
>
>
>
> > comparison with baselines is not convincing
>
> + Baseline PBT methods consume massive GPU memory for the applications we considered, so we can only adopt a population size M=4 for Table 5 due to the GPU memory limit. We remark that when M is small, even policy gradient with different random seeds (PG) seems to be a strong baseline (See our revised Table 5). Hence, the numbers in Table 4 are actually restricted in a small interval (with a lower bound equal to PG results and an upper bound equal to M=4), which makes the improvements by SIPO seemingly less significant. We would still emphasize that although the numbers are small, achieving clear improvements in these challenging applications is particularly non-trivial and has not been achieved previously in the literature.
>
> + In addition, we further conduct an experiment in the GRF 3v1 scenario with a population size M=10. Baselines include DIPG and RSPO. We present the results in Appendix B.3. Results show that SIPO clearly outperforms these baselines by consistently discovering one or more additional strategies. Please check our revised paper for details.

---

> ### Author Response · Authors · 2022-11-14
> **Author Response to Reviewer nUUy (1/2)**
>
> ### We thank the reviewer for the supportive comments and related questions. We hope our following responses can address your concerns.
>
> > Is a state-based diversity measure always more appropriate?
>
> + This is a good question. We agree with the reviewer that irrelevant features can affect the algorithm and the learned strategies may be only diverse w.r.t these features. However, we remark that the issue of irrelevant features is intrinsically challenging for general RL applications. A famous example is called the noisy TV problem, where the agent observes random noises at each step. Such an issue remains an open challenge in RL exploration and representation learning, which cannot be resolved by using action-based constraints either.
>
> + Thanks to the advantages we discussed in the paper, we generally find that state-based metrics can be preferred in challenging RL tasks. Meanwhile, we also remark that since the state dimension can be much higher than actions, it is possible that RL optimization over states may be accordingly more difficult than actions. In practice, we can design a feature selector for those most relevant features for visual diversity and run diversity learning over the filtered features.  In SMAC and GRF, we utilize the agent features (excluding enemies) as the input of diversity constraint without further modifications, as discussed in Appendix C.2. We remark that even after filtering, the agent features remain high-dimensional while our algorithm still works well. Note that using a feature selector is a common practice in many existing domains, such as novelty search [1], exploration [2], and curriculum learning [3]. There are also works studying how to extract useful low-dimensional features from observations [4, 5], which are orthogonal to our focus.
>
>
> [1] Cully, A., Clune, J., Tarapore, D., & Mouret, J. B. (2015). Robots that can adapt like animals. Nature, 521(7553), 503-507.
>
> [2] Liu, I. J., Jain, U., Yeh, R. A., & Schwing, A. (2021, July). Cooperative exploration for multi-agent deep reinforcement learning. In International Conference on Machine Learning (pp. 6826-6836). PMLR. (Long Oral)
>
> [3] Campero, A., Raileanu, R., Kuttler, H., Tenenbaum, J. B., Rocktäschel, T., & Grefenstette, E. (2020, September). Learning with AMIGo: Adversarially Motivated Intrinsic Goals. In International Conference on Learning Representations.
>
> [4] Wu, Y., Tucker, G., & Nachum, O. (2018, September). The Laplacian in RL: Learning Representations with Efficient Approximations. In International Conference on Learning Representations.
>
> [5] Ghosh, D., Gupta, A., & Levine, S. (2018, September). Learning Actionable Representations with Goal Conditioned Policies. In International Conference on Learning Representations.
>
> > But I do not see how the proposed framework can generally guarantee interpretability.
>
> + We’d like to claim that the strategies learned in GRF scenarios are human-interpretable by visually watching the emergent behaviors. However, theoretical interpretability is not guaranteed in general since how to measure the interpretability in RL is still an open challenge. We have changed relevant claims in the revised version.
>
> > How did the authors come up with the exact number (in Table 3&4)? This seems a very subjective metric, since different observers may report a different number. I was wondering whether some other methodology was followed instead.
>
> + We have made our best efforts to ensure a fair evaluation. Specifically, we summarize the evaluation criterion as follows (Appendix B.7 in our revised paper). In 3v1 and CA, players perform passes and shoot in the front yard. We consider two strategies to be different if the resulting trajectories of ball movement are different, e.g. the ball is passed to different players or different players perform a shoot. In Corner, besides ball movement, we further categorize pass-and-shoot strategies according to the position of shooting in the penalty box (e.g., lower/middle/upper spot). All the authors perform independent evaluations based on this criterion and strong agreements are achieved. Please check our project website for GIF demonstrations. We have also released all the replays we utilized for counting numbers [here](https://mega.nz/folder/xP0D2CRa#YRL-PVjjsyZhGZ2QUZqH2g) and the reviewer may check them out as well.
>
> + Moreover, similar to SMAC, we also report the state-based DvD score in GRF scenarios as a more ``objective’’ metric in Appendix B.2. Results show similar trends as Table 5.

---

### Author Response · Authors · 2022-11-14
**Common Response: Paper has been updated**

Dear reviewers, we have revised our paper according to your valuable suggestions. We have

- Added the result of naive policy gradient with random restarts in Table 5

- Added the result of PBT ablation in Table 6

- Added the result of GRF 3v1 with a population size M=10 in Appendix B.3

- Reported the state-based population diversity score of GRF in Appendix B.2

- Corrected the computation of action L2 norm in the grid-world example

- Reported the details of the evaluation process and criterion of GRF numbers in Appendix B.7

- Made a few more clarifications in the paper.

All the changes are marked in red.

---

> ### Author Response · Authors · 2022-12-04
> **Further paper update**
>
> According to the suggestions of Reviewer GEJD, we made a few updates to our draft. The updated version is **<[in this link](https://mega.nz/file/4LEwhTbb#7qeAy-fo_6-xMW9LFZutE-nUvaXfe5JauYLMarizlso)>** (as originally posted in [our reply](https://openreview.net/forum?id=OfaJyiYonBk&noteId=gJNIIEvcMnP) to Reviewer GEJD one week ago), with changes in magenta.
>
> Updates include
> 1. new Appendix B.8: To show that the Wasserstein distance metric can indeed learn effective features, we add new experiments by manually **adding random signals to the game state** (36 random dims v.s. 115 state dims). Our method still works.
> 2. We further clarify in the preliminary section that we focus on state inputs.
> 3. We add a limitation section in the conclusion section.

---

### Author Response · Authors · 2022-11-18
**A Gentle Reminder from the Authors (2022/11/18)**

Dear Reviewers:

We have updated our paper and added discussions regarding the reviewers' concerns in Appendix F to summarize the rebuttal. Since the deadline is approaching, we sincerely hope the reviewers can read our response and the revised paper. Please let us know if the reviewers have any comments about our response or any other additional concerns.

Paper 3507 Authors

---

### Decision · Program_Chairs · 2023-01-20

**Decision:**

Reject

**Justification For Why Not Higher Score:**

I am not entirely happy with the responses to reviewers nor with the treatment of related work, and the work has limited novelty.

**Justification For Why Not Lower Score:**

N/A

**Metareview: Summary, Strengths And Weaknesses:**

The paper presents a simple Iterative Learning algorithm and distance metric, and applies it to two MARL domains. The algorithm and metric are not particularly novel, but the analysis is a nice touch. However, the method might be limited to very particular environments and state representations. Two reviewers mentioned the limitation with state-based similarity measures, and did not (IMHO) get satisfying responses from the authors.

Reviewer GEJD has some valid points, particularly when it comes to how the paper is framed (a thorough analysis of existing algorithm components would have been preferably to claiming novelty and superior performance at all costs). Disappointingly, the authors seem to not really understand this reviewer's criticisms, and their answers are somewhat beside the point (for example, GEJD is not arguing that the problem with iterative learning is that big companies dont use it, which the authors seem to believe). At the same, reviewer GEJD's comments sound a little bit they wanted the authors to have written a different paper than they did, which is not a valid critique.

Still, the argument from GEJD that the algorithm would not handle procedurally generated environments is important. It's not only about the visual input, but the fact that states can be very different even for identical action sequences.

I am a little bit surprised that the authors do not mention quality-diversity in their related work, given that what their algorithm is arguably a QD algorithm. A few QD papers are cited, but erroneously referred to as "multiobjective", suggesting that the authors have not digested that literature.

On balance, I think this paper has some value despite limited novelty, but the author's somewhat disingenuous engagement with the reviewers together with the strange non-discussion of QD leads me to believe that the authors need to be engage a bit more honestly with both reviewers and related work.